# ASYMMETRIC SYNTHETIC DATA UPDATE FOR DOMAIN INCREMENTAL DATASET DISTILLATION

**Minyoung Oh & Jae-Young Sim**
Graduate School of Artificial Intelligence
Ulsan National Institute of Science and Technology (UNIST)
Ulsan, Republic of Korea
{mmyy2513,jysim}@unist.ac.kr

## ABSTRACT

Dataset distillation (DD) attempts to construct a compact synthetic dataset that serves as a proxy for a large real dataset under a fixed storage budget, thereby reducing the storage burden and training costs. Prior works assume the full dataset is available upfront which is distilled at once, although real datasets are collected incrementally over time in practice. To alleviate this gap, we introduce a new problem setting, *Domain Incremental Dataset Distillation*, that continually distills datasets from different domains into a single synthetic dataset. The conventional DD sequentially processes arriving datasets in order, overwriting the old knowledge with new one, causing catastrophic forgetting problem. To overcome this drawback, we propose *Asymmetric Synthetic Data Update* strategy that adjusts the per-sample update rates for synthetic dataset while balancing the stability-plasticity trade-off. Specifically, we design a bi-level optimization method based on meta-learning framework to estimate the optimal update rates, which allows each sample to focus on either stability or plasticity, thereby striking a balance between them. Experimental results demonstrate that our approach effectively mitigates the catastrophic forgetting and achieves superior performance of DD across continually incoming datasets compared with existing methods.

## 1 INTRODUCTION

Dataset Distillation (DD) (Wang et al., 2018) aims to generate a small synthetic dataset that encapsulates the essential information learned from a large real dataset by matching the training dynamics of models between the synthetic and real datasets. Highlighting the importance of the large scale dataset to guarantee good performance of deep learning models, DD has attracted significant attention to streamline the model training process and reduce storage costs. These advantages of DD facilitate more efficient applications in areas such as continual learning (Masarczyk & Tautkute, 2020; Yang et al., 2023; Gu et al., 2024), neural architecture search (Such et al., 2020; Medvedev & D'yakonov, 2021), and federated learning (Goetz & Tewari, 2020; Jia et al., 2024), while simultaneously addressing privacy issues (Dong et al., 2022; Chai et al., 2024; Zheng et al., 2025). Notably, a recent method (Guo et al., 2024) even surpasses the performance of the whole dataset, while requiring significantly less storage compared to the original dataset.

However, existing methods of DD only consider that the entire dataset is available in advance, even though large datasets are often collected incrementally over time in practice. In such a case, a naive approach that distills each newly arriving dataset into a seperate synthetic dataset in order, linearly increases the storage burden to accumulate the resulting synthetic datasets, as shown in Figure 1 (a). Moreover, the training costs also increases since the models should be trained on all accumulated synthetic datasets.

To address those storage and training cost issues, we formally define a new problem setting, *Domain Incremental Dataset Distillation* (DIDD), where datasets from different domains arrive sequentially while sharing the same label space. To tackle this, we introduce a novel framework that continually distills these arriving datasets into a single fixed-size synthetic dataset, as shown in Figure 1 (b). It keeps efficiency of the storage and training costs by consistently maintaining a single fixed-size

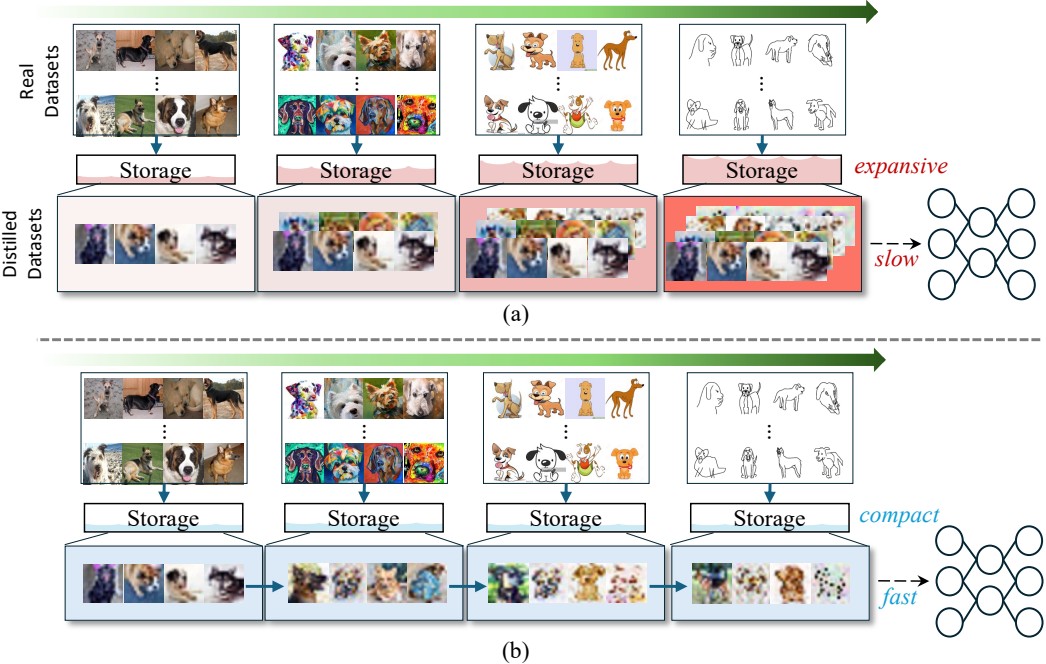

Figure 1: Comparison of (a) existing Dataset Distillation and (b) the proposed *Domain Incremental Dataset Distillation*. (a) DD considers per-dataset distillation, which increases storage and training costs depending on the number of datasets. (b) DIDD maintains a fixed-size synthetic dataset over time, thereby keeping storage and training costs comparable to those of a single dataset.

synthetic dataset over time. However, existing DD methods only focus on adapting the synthetic dataset to the most recent dataset, which is known as *plasticity*, overlooking the importance of preserving knowledge from previously seen datasets, referred to as *stability*. Consequently, the knowledge of earlier datasets in the synthetic dataset can be easily overwritten by that of the most recent one, known as *catastrophic forgetting* (McCloskey & Cohen, 1989; French, 1999).

To alleviate catastrophic forgetting, we introduce a stability loss, which preserves the knowledge of the previously seen datasets by encouraging the features of the updated synthetic dataset to remain consistent with those of the previous one. However, the stability and plasticity are inherently conflicting to each other, since the datasets collected at different time instances contain different knowledge. To mitigate this conflict, we propose an asymmetric synthetic data update strategy that adjusts per-sample update rates for the stability and plasticity objectives, respectively, using bi-level optimization within a meta-learning framework (Finn et al., 2017). In this way, each sample is encouraged to focus on either stability or plasticity, thereby achieving a balance between the two. Experimental results on Rotated MNIST (Larochelle et al., 2007), Seq-CORe50 (Lomonaco & Maltoni, 2017; Shi & Wang, 2023), and PACS (Li et al., 2017) datasets demonstrate that the proposed method effectively mitigates catastrophic forgetting, achieving superior DD performance compared with existing methods.

Our contributions are summarized as follows:

- We formally define Domain-Incremental Dataset Distillation (DIDD) as a novel problem setting and introduce a framework that distills the knowledge of continually arriving datasets into a single synthetic dataset under a fixed storage budget while preserving knowledge from previous data.

- We design an asymmetric synthetic data update strategy that adjusts per-sample update rates via bi-level optimization to effectively strike a balance between stability and plasticity.

- Experimental results demonstrate that the proposed method effectively mitigates catastrophic forgetting while accurately representing the most recent dataset, achieving superior performance compared with existing methods.

## 2 RELATED WORK

### 2.1 DATASET DISTILLATION

Dataset Distillation was first introduced by (Wang et al., 2018) and has since been actively studied to reduce both storage and training costs for large datasets. The main idea is to synthesize a compact surrogate dataset whose training dynamics induce model updates comparable to those obtained from the full dataset. Recent approaches realize this objective by matching aspects of the training dynamics between models trained on synthetic and real data. Broadly, they fall into three categories: (i) gradient matching, (ii) trajectory matching, and (iii) distribution matching. Gradient Matching (GM) methods align the parameter gradients computed on synthetic data with those computed on real data, typically by minimizing the discrepancy between the two gradient vectors (Zhao & Bilen, 2021a). Building on GM, DSA introduces differentiable data augmentation to improve the generalization of the distilled dataset (Zhao & Bilen, 2021b). Trajectory Matching (TM) methods encourage the optimization path of parameters trained on the synthetic data to follow the path observed when training on the real data (Cazenavette et al., 2022). Recent TM-based methods achieve superior performance by analyzing the knowledge contained in the early or late segments of the trajectory (Guo et al., 2024) or by emphasizing class-discriminative regions during image updates (Wang et al., 2025). However, they typically require substantial computation, as distillation involves unrolling and backpropagating through multiple optimization steps. Distribution Matching (DM) methods align feature distributions computed from synthetic and real data (Zhao & Bilen, 2023) and are generally the most computationally efficient among the three categories. Recent studies leverage intermediate features for alignment (Wang et al., 2022) and develop improved distance metrics to measure distributional similarity (Zhang et al., 2024), leading to more faithful distilled datasets. However, existing methods usually assume that the entire dataset is available upfront and can be revisited repeatedly, which is often unrealistic in practice. As a result, they do not directly address scenarios where datasets arrive sequentially over time under a fixed storage budget.

### 2.2 DOMAIN INCREMENTAL LEARNING

Domain Incremental Learning (DIL) is a subfield of continual learning (Van de Ven et al., 2022) that considers settings where datasets from different domains arrive sequentially while sharing the same label space. In this scenario, the set of classes remains fixed, while the data distribution shifts across domains due to differences in acquisition conditions, sensors, styles, or backgrounds. Such domain shifts often lead to catastrophic forgetting, where knowledge acquired on earlier domains is overwritten by that of recent domains. Approaches to DIL are commonly grouped into three categories: (i) regularization-based, (ii) replay-based, and (iii) dynamic architecture-based methods. Regularization-based methods constrain updates on parameters deemed important for previous domains so that those parameters do not change excessively. For example, EWC estimates parameter importance using the Fisher information (Kirkpatrick et al., 2017), while MAS leverages parameter sensitivity to achieve a similar effect (Aljundi et al., 2018). Replay-based methods store a subset of past data and train with it alongside current data to resist distribution shift. GEM imposes constraints that prevent gradients from harming past performance (Lopez-Paz & Ranzato, 2017), whereas ER mixes exemplars from past and present domains directly in minibatches (Rolnick et al., 2019). Dynamic architecture-based methods expand model capacity when new domains arrive, helping to isolate knowledge and reduce interference. PNN grows the architecture roughly linearly with the number of domains (Rusu et al., 2016), while other approaches expand capacity adaptively to balance sharing and specialization (Hung et al., 2019). Despite these advances, it remains underexplored how to mitigate catastrophic forgetting in DD when heterogeneous domain datasets continue to arrive.

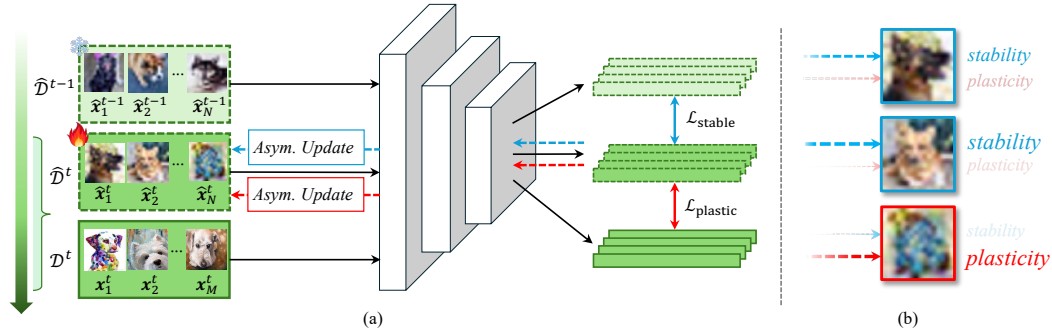

Figure 2: Overview of (a) the proposed framework for Domain Incremental Dataset Distillation utilizing (b) the asymmetric synthetic data update strategy.

## 3 PRELIMINARY

The conventional dataset distillation aims to synthesize a small dataset $\hat{\mathcal{D}}$ which serves as a proxy for a large real dataset $\mathcal{D}$ where $|\hat{\mathcal{D}}| \ll |\mathcal{D}|$, such that the models trained on $\hat{\mathcal{D}}$ achieves comparable performance to those trained on $\mathcal{D}$. Concretely, the goal of DD can be formulated as follows:

$$\min_{\hat{\mathcal{D}}} \mathbb{E}_{\boldsymbol{x} \sim P(\mathcal{D}^*)} \left[ \mathcal{L}\big(\phi_{\hat{\mathcal{D}}}(\boldsymbol{x}), \phi_{\mathcal{D}}(\boldsymbol{x})\big) \right], \tag{1}$$

where $P(\mathcal{D}^*)$ denotes target data distribution, $\mathcal{L}(a, b)$ measures the difference between $a$ and $b$, and $\phi_{\hat{\mathcal{D}}}$ and $\phi_{\mathcal{D}}$ indicate the models trained on $\hat{\mathcal{D}}$ and $\mathcal{D}$, respectively. To achieve this goal, recent methods synthesize $\hat{\mathcal{D}}$ by matching training dynamics (i.e., gradient, trajectory, and feature distribution) of models trained on $\hat{\mathcal{D}}$ and $\mathcal{D}$, respectively. In this paper, we focus on the DM method (Zhao & Bilen, 2023), which matches the feature distributions of $\hat{\mathcal{D}}$ to those of $\mathcal{D}$ in a class-wise manner, computed using a randomly initialized model $\phi$.

## 4 METHODOLOGY

Figure 2 illustrates the proposed Domain Incremental Dataset Distillation framework, which asymmetrically updates the per-sample rates. This strategy balances stability and plasticity, allowing each image to prioritize one over the other.

### 4.1 PROBLEM FORMULATION

Conventional dataset distillation generates a synthetic dataset from each dataset independently, assuming that the entire dataset $\mathcal{D}$ is fully available in advance. However, when sequential datasets $\{\mathcal{D}^t\}_{t=1}^{T}$ arrive over time, such per-dataset distillation scheme increases the storage and training costs. To overcome these drawbacks, we formally define a new problem setting, Domain Incremental Dataset Distillation, where datasets from different domains arrive sequentially while sharing the same label space. To address this problem, we propose a novel framework that continually distills these datasets into a single synthetic dataset. Specifically, let the dataset $\mathcal{D}^t$ with $C$ classes arrives at the $t$-th time instance. The goal of DIDD is to synthesize an optimal dataset $\hat{\mathcal{D}}^t$ that not only contains the knowledge of the current dataset $\mathcal{D}^t$ but preserves the knowledge of the previous datasets, $\mathcal{D}^{1:t-1}$, subject to a contraint of a fixed storage budget, $|\hat{\mathcal{D}}^t| = \text{IPC} \times C$ where IPC means the Images Per Class. Consequently, we formulate the problem of DIDD as follows:

$$\min_{\hat{\mathcal{D}}^t} \mathbb{E}_{\boldsymbol{x} \sim P(\mathcal{D}^*)} \left[ \mathcal{L}\big(\phi_{\hat{\mathcal{D}}^t}(\boldsymbol{x}), \phi_{\mathcal{D}^{1:t}}(\boldsymbol{x})\big) \right] \quad s.t. \ |\hat{\mathcal{D}}^t| = \text{IPC} \times C. \tag{2}$$

Note that DIDD does not incur increasing storage or training costs over time, since it consistently maintains a single fixed-size synthetic dataset $\hat{\mathcal{D}}^t$.

## 4.2 Asymmetric Update via Bi-level Optimization

**Plasticity and Stability.** We construct a baseline following (Zhang et al., 2024), minimizing the Maximum Mean Discrepancy (MMD) between two feature distributions of $\mathcal{D}^t$ and $\hat{\mathcal{D}}^t$, defined as the plasticity loss:

$$\mathcal{L}_{\text{plastic}}(\hat{\boldsymbol{x}}_c^t) = d(F(\boldsymbol{x}_c^t), F(\hat{\boldsymbol{x}}_c^t)). \tag{3}$$

Here $d(A, B)$ is the MMD with the Gaussian kernel between two sets of $A$ and $B$. $F(\boldsymbol{x}_c^t) = \{\phi(\boldsymbol{x}_{c,i}^t)\}_{i=1}^M$ and $F(\hat{\boldsymbol{x}}_c^t) = \{\phi(\hat{\boldsymbol{x}}_{c,i}^t)\}_{i=1}^N$, where $\boldsymbol{x}_{c,i}^t$ and $\hat{\boldsymbol{x}}_{c,i}^t$ indicate the $i$-th image of class $c$ in $\mathcal{D}^t$ and $\hat{\mathcal{D}}^t$, with $N \ll M$. We omit the class index $c$ throughout this paper for simplicity. Minimizing $\mathcal{L}_{\text{plastic}}$ increases plasticity $\mathcal{P}$ making $\hat{\mathcal{D}}$ a faithful proxy for $\mathcal{D}$.

However, existing DD methods focus solely on maximizing $\mathcal{P}$ adapting to given datasets. Therefore, directly applying them to the DIDD setting results in $\hat{\mathcal{D}}^t$ only containing the knowledge of $\mathcal{D}^t$, while forgetting that of $\hat{\mathcal{D}}^{t-1}$, which is known as catastrophic forgetting (McCloskey & Cohen, 1989; French, 1999). To mitigate this problem, it is essential to promote the stability $\mathcal{S}$ of the synthetic dataset. Thus, we introduce a stability loss $\mathcal{L}_{\text{stable}}$ that matches the features of $\hat{\mathcal{D}}^t$ to those of $\hat{\mathcal{D}}^{t-1}$

$$\mathcal{L}_{\text{stable}}(\hat{\boldsymbol{x}}^t) = d(F(\hat{\boldsymbol{x}}^{t-1}), F(\hat{\boldsymbol{x}}^t)). \tag{4}$$

Minimizing $\mathcal{L}_{\text{stable}}$ increases $\mathcal{S}$ of $\hat{\mathcal{D}}^t$ by regularizing its features to remain consistent with those of $\hat{\mathcal{D}}^{t-1}$, thereby preserving knowledge from earlier datasets.

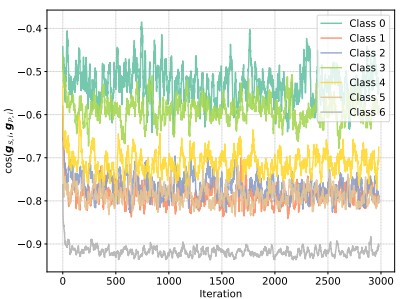

Figure 3: Cosine similarity between $\boldsymbol{g}_{\mathcal{S},i}$ and $\boldsymbol{g}_{\mathcal{P},i}$ on PACS.

By jointly minimizing $\mathcal{L}_{\text{plastic}}$ and $\mathcal{L}_{\text{stable}}$, each image $\hat{\boldsymbol{x}}_i^t$ is updated while considering both plasticity and stability simultaneously, formulated as:

$$\hat{\boldsymbol{x}}_i^t \leftarrow \hat{\boldsymbol{x}}_i^t - \eta_x(\boldsymbol{g}_{\mathcal{S},i} + \boldsymbol{g}_{\mathcal{P},i}), \tag{5}$$

where $\eta_x$ is the learning rate for the synthetic data and

$$\boldsymbol{g}_{\mathcal{S},i} = \frac{\partial \mathcal{L}_{\text{stable}}(\hat{\boldsymbol{x}}^t)}{\partial \hat{\boldsymbol{x}}_i^t}, \qquad \boldsymbol{g}_{\mathcal{P},i} = \frac{\partial \mathcal{L}_{\text{plastic}}(\hat{\boldsymbol{x}}^t)}{\partial \hat{\boldsymbol{x}}_i^t}. \tag{6}$$

However, as shown in Figure 3, $\boldsymbol{g}_{\mathcal{S},i}$ and $\boldsymbol{g}_{\mathcal{P},i}$ often conflict with each other, as indicated by their negative cosine similarity. This is because $\boldsymbol{g}_{\mathcal{S},i}$ preserves knowledge from $\hat{\mathcal{D}}^{t-1}$ while $\boldsymbol{g}_{\mathcal{P},i}$ adapts to $\mathcal{D}^t$, leading to conflicting objectives. Therefore, applying the same update rate to $\boldsymbol{g}_{\mathcal{S},i}$ and $\boldsymbol{g}_{\mathcal{P},i}$ is potentially suboptimal.

**Asymmetric Synthetic Data Update.** To address this issue, we propose an asymmetric synthetic data update strategy that assigns distinct update rates to $\boldsymbol{g}_{\mathcal{S},i}$ and $\boldsymbol{g}_{\mathcal{P},i}$ for each sample $\hat{\boldsymbol{x}}_i^t$. Specifically, we introduce scale parameters $\boldsymbol{\alpha} = \{\alpha_i\}_{i=1}^N$ and $\boldsymbol{\beta} = \{\beta_i\}_{i=1}^N$ to scale $\boldsymbol{g}_{\mathcal{S},i}$ and $\boldsymbol{g}_{\mathcal{P},i}$, respectively. Accordingly, Eq. 5 is modified as

$$\hat{\boldsymbol{x}}_i^t \leftarrow \hat{\boldsymbol{x}}_i^t - \eta_x(\bar{\alpha}_i \cdot \boldsymbol{g}_{\mathcal{S},i} + \bar{\beta}_i \cdot \boldsymbol{g}_{\mathcal{P},i}). \tag{7}$$

Here $\bar{\alpha}_i$ and $\bar{\beta}_i$ are bounded values by sigmoid functions within $(\alpha_{\min}, \alpha_{\max})$ and $(\beta_{\min}, \beta_{\max})$, respectively, which are formulated as:

$$\bar{\alpha}_i = \alpha_{\min} + \frac{\alpha_{\max} - \alpha_{\min}}{1 + \exp(-\alpha_i)}, \quad \bar{\beta}_i = \beta_{\min} + \frac{\beta_{\max} - \beta_{\min}}{1 + \exp(-\beta_i)}. \tag{8}$$

This asymmetric update strategy enables each sample to prioritize either stability or plasticity by adaptively adjusting $\bar{\alpha}_i$ and $\bar{\beta}_i$. For instance, $\hat{\boldsymbol{x}}_i^t$ emphasizes stability over plasticity if $\bar{\alpha}_i > \bar{\beta}_i$, and it emphasizes plasticity if $\bar{\alpha}_i < \bar{\beta}_i$. In this way, the conflict between stability and plasticity can be mitigated.

However, it is not trivial to determine the optimal values of $\alpha_i$ and $\beta_i$ for each sample. To this end, we propose a bi-level optimization method based on the meta-learning framework (Finn et al., 2017), treating that treats $\alpha_i$ and $\beta_i$ as learnable parameters. Here $\alpha_i$ and $\beta_i$ are initialized as 0. Specifically, to find the optimal $\alpha_i$ and $\beta_i$, we first perform a meta-update of $\hat{\boldsymbol{x}}_i^t$ by using Eq. 7:

$$\hat{\boldsymbol{x}}_{\text{meta},i}^t = \hat{\boldsymbol{x}}_i^t - \eta_x(\bar{\alpha}_i \cdot \boldsymbol{g}_{\mathcal{S},i} + \bar{\beta}_i \cdot \boldsymbol{g}_{\mathcal{P},i}). \tag{9}$$

Using $\hat{\boldsymbol{x}}_{\text{meta},i}^t$, we compute the meta-loss $\mathcal{L}_{\text{meta}}(\hat{\boldsymbol{x}}_{\text{meta}}^t)$:

$$\mathcal{L}_{\text{meta}}(\hat{\boldsymbol{x}}_{\text{meta}}^t) = \mathcal{L}_{\text{stable}}(\hat{\boldsymbol{x}}_{\text{meta}}^t) + \mathcal{L}_{\text{plastic}}(\hat{\boldsymbol{x}}_{\text{meta}}^t). \tag{10}$$

Minimizing $\mathcal{L}_{\text{meta}}$ guides the search for optimal $\boldsymbol{\alpha}$ and $\boldsymbol{\beta}$ that maximize $\mathcal{S}$ and $\mathcal{P}$, respectively.

However, this process may lead to a trivial solution in which both $\bar{\alpha}_i$ and $\bar{\beta}_i$ are maximized indiscriminately. To analyze this, we consider the first-order Taylor expansion of $\mathcal{L}_{\text{meta}}(\hat{\boldsymbol{x}}_{\text{meta}}^t)$ around $\hat{\boldsymbol{x}}^t$:

$$\mathcal{L}_{\text{meta}}(\hat{\boldsymbol{x}}_{\text{meta}}^t) \approx \mathcal{L}_{\text{meta}}(\hat{\boldsymbol{x}}^t) - \eta_x \left\langle \frac{\partial}{\partial \hat{\boldsymbol{x}}^t} \mathcal{L}_{\text{meta}}(\hat{\boldsymbol{x}}^t), \sum_{i=1}^{N} \left( \bar{\alpha}_i \cdot \boldsymbol{g}_{\mathcal{S},i} + \bar{\beta}_i \cdot \boldsymbol{g}_{\mathcal{P},i} \right) \right\rangle. \tag{11}$$

This implies that large values of $\bar{\alpha}_i$ and $\bar{\beta}_i$ for all $i$ can significantly decrease in $\mathcal{L}_{\text{meta}}(\hat{\boldsymbol{x}}_{\text{meta}}^t)$ if the inner products $\langle \frac{\partial}{\partial \hat{\boldsymbol{x}}^t} \mathcal{L}_{\text{meta}}(\hat{\boldsymbol{x}}^t), \boldsymbol{g}_{\mathcal{S},i} \rangle$ and $\langle \frac{\partial}{\partial \hat{\boldsymbol{x}}^t} \mathcal{L}_{\text{meta}}(\hat{\boldsymbol{x}}^t), \boldsymbol{g}_{\mathcal{P},i} \rangle$ are positive. To prevent this trivial solution and enforce selectivity, we penalize the magnitude of the average per-sample update rates, $\bar{\alpha}_i$ and $\bar{\beta}_i$, by minimizing:

$$\mathcal{L}_{\text{penalty}-\alpha} = \frac{1}{N} \sum_{i=1}^{N} \bar{\alpha}_i, \quad \mathcal{L}_{\text{penalty}-\beta} = \frac{1}{N} \sum_{i=1}^{N} \bar{\beta}_i. \tag{12}$$

This regularization discourages $\bar{\alpha}_i$ and $\bar{\beta}_i$ from growing indiscriminately, thereby encouraging an asymmetric update strategy.

Consequently, we formulate the overall meta-objective as:

$$\mathcal{L}_{\text{meta}}^{\text{penalty}} = \mathcal{L}_{\text{meta}}(\hat{\boldsymbol{x}}_{\text{meta}}^t) + \lambda_\alpha \mathcal{L}_{\text{penalty}-\alpha} + \lambda_\beta \mathcal{L}_{\text{penalty}-\beta}. \tag{13}$$

By minimizing $\mathcal{L}_{\text{meta}}^{\text{penalty}}$, we update the learnable parameters $\alpha_i$ and $\beta_i$ as follows:

$$\alpha_i \leftarrow \alpha_i - \eta_\alpha \frac{\partial}{\partial \alpha_i} \mathcal{L}_{\text{meta}}^{\text{penalty}}, \quad \beta_i \leftarrow \beta_i - \eta_\beta \frac{\partial}{\partial \beta_i} \mathcal{L}_{\text{meta}}^{\text{penalty}}, \tag{14}$$

where $\eta_\alpha$ and $\eta_\beta$ are the learning rates for $\alpha_i$ and $\beta_i$, and $\lambda_\alpha$ and $\lambda_\beta$ are hyperparameters. In this way, we obtain $\boldsymbol{\alpha}$ and $\boldsymbol{\beta}$ that maximize $\mathcal{S}$ and $\mathcal{P}$, respectively, while keeping their average magnitudes minimal. Finally, updating $\hat{\boldsymbol{x}}$ with the learned $\boldsymbol{\alpha}$ and $\boldsymbol{\beta}$ encourages each sample to specialize in either $\mathcal{S}$ or $\mathcal{P}$, yielding a synthetic dataset $\hat{\mathcal{D}}^t$ that strikes an optimal balance between them.

**Theoretical Interpretation via KKT Conditions.** We view DIDD as a constrained optimization problem in which the synthetic data $\hat{\boldsymbol{x}}^t$ is required to satisfy stability and plasticity requirements. Let $\epsilon_{\text{stable},i}$ and $\epsilon_{\text{plastic},i}$ denote the desired tolerance thresholds for each sample $\hat{\boldsymbol{x}}_i^t$. Conceptually, the primal problem can be written as

$$\text{find } \hat{\boldsymbol{x}}^t \quad \text{s.t.} \quad \mathcal{L}_{\text{stable}}(\hat{\boldsymbol{x}}_i^t) \leq \epsilon_{\text{stable},i}, \quad \mathcal{L}_{\text{plastic}}(\hat{\boldsymbol{x}}_i^t) \leq \epsilon_{\text{plastic},i}, \quad \forall i. \tag{15}$$

Here the thresholds $\epsilon_{\text{stable},i}$ and $\epsilon_{\text{plastic},i}$ describe the desired upper bounds on the two losses.

Introducing non-negative Lagrange multipliers $\boldsymbol{\alpha}, \boldsymbol{\beta} \in \mathbb{R}_{\geq 0}^N$, we obtain the Lagrangian function:

$$\mathcal{L}(\hat{\boldsymbol{x}}^t, \boldsymbol{\alpha}, \boldsymbol{\beta}) = \sum_{i=1}^{N} \left[ \bar{\alpha}_i \left( \mathcal{L}_{\text{stable}}(\hat{\boldsymbol{x}}_i^t) - \epsilon_{\text{stable},i} \right) + \bar{\beta}_i \left( \mathcal{L}_{\text{plastic}}(\hat{\boldsymbol{x}}_i^t) - \epsilon_{\text{plastic},i} \right) \right], \tag{16}$$

where $\bar{\alpha}_i$ and $\bar{\beta}_i$ are the bounded versions of $\alpha_i$ and $\beta_i$ defined in Eq. 8. Note that the gradient of $\mathcal{L}$ with respect to $\hat{\boldsymbol{x}}_i^t$ aligns with the proposed asymmetric update direction in Eq. 7. This provides an interpretation of $\bar{\alpha}_i$ and $\bar{\beta}_i$ as quantities analogous to sample-wise Lagrange multipliers, controlling the extent to which each constraint influences the update of $\hat{\boldsymbol{x}}_i^t$.

In the ideal constrained problem, the KKT conditions imply the complementary slackness relations

$$\bar{\alpha}_i \left( \mathcal{L}_{\text{stable}}(\hat{\boldsymbol{x}}_i^t) - \epsilon_{\text{stable},i} \right) = 0, \quad \bar{\beta}_i \left( \mathcal{L}_{\text{plastic}}(\hat{\boldsymbol{x}}_i^t) - \epsilon_{\text{plastic},i} \right) = 0, \tag{17}$$

meaning that the multipliers become non-zero only when the corresponding constraints are active. Our meta-learning objective approximates this behavior by optimizing $\boldsymbol{\alpha}$ and $\boldsymbol{\beta}$ that enforces the samples with relatively close to violating stability or plasticity constraints to receive larger update rates. Therefore, the KKT formulation serves as an interpretive lens that connects the asymmetric update rule to a constrained optimization perspective and clarifies the role of $\boldsymbol{\alpha}$ and $\boldsymbol{\beta}$ as stability and plasticity weights.

## 5 EXPERIMENTS

### 5.1 EXPERIMENTAL SETUP

#### 5.1.1 DATASETS

To verify the effectiveness of our method, we use three datasets: Rotated MNIST (R-MNIST) (Larochelle et al., 2007), Seq-CORe50 (Lomonaco & Maltoni, 2017; Shi & Wang, 2023), and PACS (Li et al., 2017). R-MNIST is a variant of MNIST with 20 sequential domains, each containing the 10 digit classes; the images in each domain are rotated by a domain specific angle that increases across domains. Seq-CORe50 contains 50 object classes collected over 11 different locations, which are treated as distinct domains. PACS contains 7 classes from 4 domains: Photo, Art painting, Cartoon, and Sketch. We describe more details about datasets in Appendix A.2.

#### 5.1.2 IMPLEMENTATION DETAILS

All experiments are implemented with PyTorch and run on a single NVIDIA RTX 3090 GPU. For all datasets, images are resized to $32 \times 32$ and a simple 3 layer ConvNet is used as the feature extractor with instance normalization for distillation, following prior DD work. For distillation, we use the SGD optimizer and run 3,000 iterations per domain. We apply data augmentations including ColorJitter, RandomCrop, and Cutout (DeVries & Taylor, 2017) following (Zhao & Bilen, 2021b), and the multi formation function following (Kim et al., 2022; Zhang et al., 2024). Hyperparameters are set as $\alpha_{\min} = \beta_{\min} = 0$, $\alpha_{\max} = \beta_{\max} = 2$, $\eta_\alpha = \eta_\beta = $ 1e-2, and $\lambda_\alpha = \lambda_\beta = $ 1e-4. More detailed hyperparameters are provided in the Appendix A.5. After distillation, we evaluate the distilled dataset by training a model from scratch for 1,000 epochs using SGD with learning rate 1e-2 and batch size 64.

#### 5.1.3 METRICS

After distillation from the $T$-th domain dataset, we train a network on $\hat{\mathcal{D}}^T$ and evaluate the distilled dataset using two metrics, average accuracy $\mathcal{A}^T$ and average forgetting $\mathcal{F}^T$. The average accuracy is the mean of per domain accuracies after distillation of $\mathcal{D}^T$:

$$\mathcal{A}^T = \frac{1}{T} \sum_{t=1}^{T} a_t^T, \tag{18}$$

where $a_t^k$ denotes the accuracy on the $t$-th domain, evaluated with the model trained on $\hat{\mathcal{D}}^k$ after distillation from the $k$-th domain dataset. The average forgetting $\mathcal{F}^T$ measures the mean drop from the best accuracy previously achieved on each domain to its accuracy after the $T$-th domain dataset distillation:

$$\mathcal{F}^T = \frac{1}{T-1} \sum_{t=1}^{T-1} \left( \max_{1 \leq k < T} a_t^k - a_t^T \right). \tag{19}$$

All results are reported as the mean and standard deviation over three runs.

### 5.2 PERFORMANCE COMPARISON

We compare our method on three datasets when IPC is 1, 10, and 20 on R-MNIST, and 10 and 20 on Seq-CORe50 and PACS, using the following baselines (i) *Whole dataset*: training on the entire real datasets across all domains, where the storage budget is $\sum_{i=1}^{T} |\mathcal{D}^t|$. (ii) *Distill-Gather (Gather)*: training on the union of per domain distilled datasets as conventional DD methods (as shown in Figure 1 (a)), where the storage budget is $\sum_{i=1}^{T} |\hat{\mathcal{D}}^t|$. (iii) *Joint*: training on a dataset distilled jointly from all domains, where the storage budget is $|\hat{\mathcal{D}}^T|$, thereby serving as an upper bound. (iv) *Fine-tune*: training on a dataset distilled sequentially from each domain without additional techniques. For more diverse baseline construction, we additionally implement *Gather* and *Joint* methods based on DSA (Zhao & Bilen, 2021b) and DC (Zhao & Bilen, 2021a), which are most widely used DD methods.

Since this is the first study on domain incremental dataset distillation, there is no existing method for direct comparison. Therefore, we also compare several regularization-based DIL methods, LwF (Li

| Dataset | R-MNIST | | | | | | Seq-CORe50 | | | | PACS | | | |
|---|---|---|---|---|---|---|---|---|---|---|---|---|---|---|
| IPC | 1 | | 10 | | 20 | | 10 | | 20 | | 10 | | 20 | |
| Metric | $\mathcal{A}^T$ (↑) | $\mathcal{F}^T$ (↓) | $\mathcal{A}^T$ (↑) | $\mathcal{F}^T$ (↓) | $\mathcal{A}^T$ (↑) | $\mathcal{F}^T$ (↓) | $\mathcal{A}^T$ (↑) | $\mathcal{F}^T$ (↓) | $\mathcal{A}^T$ (↑) | $\mathcal{F}^T$ (↓) | $\mathcal{A}^T$ (↑) | $\mathcal{F}^T$ (↓) | $\mathcal{A}^T$ (↑) | $\mathcal{F}^T$ (↓) |
| Real Dataset (Budget: $\sum_{i=1}^{T}|\mathcal{D}^t|$) | | | | | | | | | | | | | | |
| Wholedata | 98.9±0.1 | - | 98.9±0.1 | - | 98.9±0.1 | - | 98.6±0.3 | - | 98.6±0.3 | - | 66.1±3.6 | - | 66.1±3.6 | - |
| Conventional Dataset Distillation (Budget: $\sum_{i=1}^{T}|\hat{\mathcal{D}}^t|$) | | | | | | | | | | | | | | |
| Gather (M3D) | 92.8±0.2 | - | 94.0±0.2 | - | 93.9±0.1 | - | 96.5±0.3 | - | 97.5±0.3 | - | 56.8±1.6 | - | 59.5±0.4 | - |
| Gather (DSA) | 74.4±1.1 | - | 92.9±0.9 | - | 94.4±1.0 | - | 69.9±0.9 | - | 76.0±0.4 | - | 50.2±1.3 | - | 54.1±1.1 | - |
| Gather (DC) | 74.0±0.4 | - | 87.8±0.6 | - | 94.4±0.5 | - | 71.0±0.8 | - | 80.1±0.6 | - | 50.4±0.4 | - | 52.5±0.3 | - |
| Joint Dataset Distillation (Budget: $|\hat{\mathcal{D}}^t|$) | | | | | | | | | | | | | | |
| Joint (M3D) | 80.2±0.8 | - | 90.9±0.4 | - | 91.5±0.3 | - | 76.3±0.2 | - | 84.7±0.2 | - | 52.7±0.4 | - | 54.5±1.6 | - |
| Joint (DSA) | 55.4±1.0 | - | 82.3±0.4 | - | 86.4±0.5 | - | 33.2±0.7 | - | 41.2±0.7 | - | 46.1±1.0 | - | 48.6±1.0 | - |
| Joint (DC) | 53.7±1.0 | - | 83.6±0.4 | - | 87.4±0.4 | - | 34.7±0.5 | - | 42.4±0.9 | - | 45.3±0.9 | - | 48.4±0.1 | - |
| Domain Incremental Dataset Distillation (Budget: $|\hat{\mathcal{D}}^t|$) | | | | | | | | | | | | | | |
| Finetune | 38.9±1.6 | 59.2±1.6 | 39.4±2.2 | 59.7±2.0 | 41.9±2.1 | 56.7±2.1 | 25.4±0.7 | 79.9±0.7 | 26.4±0.2 | 79.2±0.1 | 26.9±0.8 | 44.3±0.3 | 27.4±1.2 | 44.4±1.9 |
| LwF | 32.9±3.3 | 48.5±1.8 | 32.8±3.6 | 44.0±2.9 | 36.2±3.2 | **38.7±2.0** | 16.2±0.2 | **25.3±0.9** | 17.6±0.3 | **24.7±0.2** | 35.0±0.6 | 13.0±0.3 | 34.5±2.1 | 14.5±2.2 |
| EWC | 38.8±0.1 | 59.2±0.3 | 40.7±1.9 | 58.4±1.9 | 41.6±2.3 | 56.7±2.3 | 25.4±0.4 | 79.6±0.4 | 26.4±0.2 | 79.1±0.4 | 26.4±0.7 | 43.7±0.5 | 28.6±1.5 | 43.5±1.9 |
| LF | 39.0±1.0 | 59.3±1.0 | 42.6±0.4 | 56.3±0.5 | 42.9±1.6 | 55.0±1.6 | 25.7±0.1 | 79.0±0.0 | 25.9±0.2 | 79.7±0.4 | 27.4±3.4 | 42.9±2.4 | 28.5±1.2 | 44.0±1.4 |
| MAS | 43.2±1.5 | 50.2±1.7 | 45.6±0.9 | 56.3±0.8 | 45.0±3.0 | 51.8±3.4 | 26.4±0.4 | 60.3±0.2 | 27.0±0.4 | 66.1±0.5 | 35.0±1.7 | **7.9±1.0** | 35.6±1.3 | 12.9±1.3 |
| Proposed | **58.6±1.3** | **21.0±0.8** | **62.7±2.0** | **34.9±2.0** | **59.0±0.5** | 39.3±0.4 | **55.4±0.7** | 38.8±0.5 | **60.6±0.4** | 38.7±0.3 | **48.0±3.1** | 11.2±1.8 | **52.1±0.6** | **10.0±0.5** |

Table 1: Performance Comparison on R-MNIST, Seq-CORe50, and PACS datasets.

& Hoiem, 2017), EWC (Kirkpatrick et al., 2017), MAS (Aljundi et al., 2018), and LF (Jung et al., 2016), by substituting the regularization target from model parameters to synthetic data. Note that we do not consider other methods that require extra memory such as replay and dynamic architecture-based methods, which conflicts with the goal of DD to minimize storage.

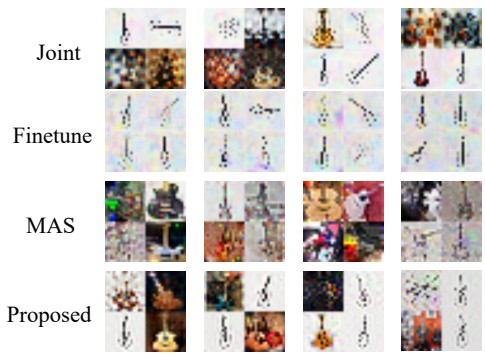

Figure 4: Synthesized images (guitar) on PACS.

Table 1 reports performance on R-MNIST, Seq-CORe50, and PACS, respectively. Using all real datasets across domains, Wholedata achieves the best performance as expected, but it incurs the largest storage and training costs. Gather is second best because it uses all per domain distilled datasets, however it still suffers from increasing storage and training costs as the number of domains grows. Joint can be regarded as an upper bound for the DIDD task since it maintains storage and training efficiency comparable to a single synthetic dataset by distilling from all domains jointly. Finetune serves as a lower bound and performs the worst in most cases due to catastrophic forgetting. We observe that EWC and LF fail to preserve knowledge from earlier datasets, resulting in performance similar to Finetune. MAS and LwF both perform better on PACS, which has a relatively short sequence, but LwF degrades on the longer sequence datasets R-MNIST and Seq-CORe50, whereas MAS is only slightly better than Finetune on those datasets. Although some methods report the lowest $\mathcal{F}^T$ in certain cases, this largely results from low overall accuracy, which reduces forgetting scores under Eq. 19. In contrast, our method outperforms Finetune and the other methods by a clear margin across all settings. Remarkably, the proposed method surpasses even the Joint upper bounds of established baselines (DSA and DC) on R-MNIST (at IPC = 1), Seq-CORe50 and PACS (at IPC = 10, 20). In particular, on the PACS dataset, our method recovers over 91% and 95% of the Joint performance at IPC = 10 and IPC = 20, respectively, validating its high data efficiency.

We also visualize synthesized images on PACS in Figure 4. Images synthesized by the Joint method show that multiple domains are well captured together, whereas Finetune mainly reflects the most recent domain, Sketch. MAS produces images biased toward earlier domains such as Photo and Art painting, indicating poor plasticity. By contrast, images synthesized by our method closely resemble those of Joint, which verifies that our method effectively strikes a balance between stability and plasticity. More qualtitative results are provided in Appendix A.9.

Figure 5 shows the average accuracy as $t$ increases on each dataset for IPC = 10 and IPC = 20. Baselines including Finetune and other continual learning methods exhibit substantial degradation as $t$ grows due to catastrophic forgetting. Meanwhile, our method preserves knowledge from earlier datasets and achieves the best performance across $t$ on all datasets.

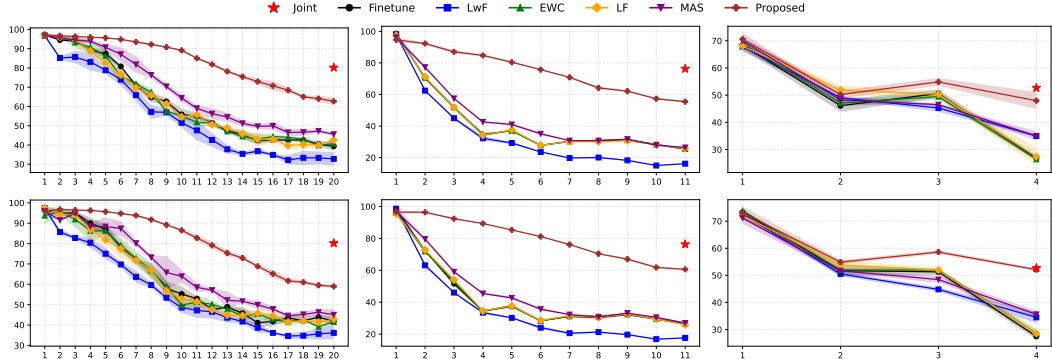

Figure 5: $\mathcal{A}^t$ across $t$ on R-MNIST (left), Seq-CORe50 (middle), and PACS (right) when IPC = 10 (first row) and IPC = 20 (second row).

**Distillation Cost Analysis.** In Table 2, we analyzed the distillation cost of the proposed method compared to that of other baselines, evaluated on the PACS dataset when IPC=10. The proposed method incurs a higher distillation cost compared to other baselines due to the additional process of bi-level optimization.

| Method | Time (s) | | | | | Performance (%) | |
|---|---|---|---|---|---|---|---|
| | P | A | C | S | Total | $\mathcal{A}^T$ ($\uparrow$) | $\mathcal{F}^T$ ($\downarrow$) |
| Finetune | 112.2 | 102.1 | 101.9 | 94.6 | 410.7 | 26.9±0.8 | 44.3±0.3 |
| LwF | 109.4 | 145.2 | 130.4 | 139.0 | 524.0 | 35.0±0.6 | 13.0±0.3 |
| EWC | 112.5 | 121.9 | 109.8 | 114.1 | 458.3 | 26.4±0.7 | 43.7±0.5 |
| LF | 103.8 | 151.0 | 143.9 | 141.4 | 540.0 | 27.4±3.4 | 42.9±2.4 |
| MAS | 107.5 | 117.3 | 103.6 | 97.5 | 425.8 | 35.0±1.7 | **7.9**±1.0 |
| Proposed | 111.4 | 342.5 | 334.2 | 335.1 | 1,123.3 | **48.0**±3.1 | 11.2±1.8 |
| Proposed (1,000 iter) | 37.5 | 109.6 | 99.6 | 97.4 | 344.1 | 46.3±1.8 | 8.5±1.2 |

Table 2: Distillation cost analysis on PACS under IPC=10.

However, it is important to note that DD is fundamentally an offline procedure executed only once prior to model training, and thus does not incur the complexity associated with real-time processing. In addition, we conducted an experiment to scale the distillation cost by adjusting the number of iterations as 1,000 during the distillation process. Note that, even with a smaller number of iterations, the proposed method (Proposed (1,000 iter)) still outperforms other baselines significantly, while achieving the lowest distillation cost.

## 5.3 ABLATION STUDY

To assess the effectiveness of our method, we conduct ablation studies on R-MNIST, which has the longest sequence among the three datasets. To examine each component with respect to the stability-plasticity trade-off, we split R-MNIST into four segments of five sequential domains $\left( \mathcal{D}^{1:5}, \mathcal{D}^{6:10}, \mathcal{D}^{11:15}, \mathcal{D}^{16:20} \right)$ and analyze the average accuracy on each segment. Table 3 reports the results for IPC = 10 and IPC = 20. We denote by $\mathcal{A}^T_{1:5}$ the average accuracy on $\mathcal{D}^{1:5}$ after distillation of the $T$-th domain, where $\mathcal{A}^T_{i:j} = \frac{1}{j-i+1} \sum_{t=i}^{j} a^T_t$. We observe that the Finetune performs poorly on earlier segments due to catastrophic forgetting. Adding $\mathcal{L}_{\text{stable}}$ improves performance on recent segments, but the earliest segment $\mathcal{A}^T_{1:5}$ degrades. This suggests that when the sequence length $T$ is large, $\mathcal{L}_{\text{stable}}$ mainly preserves recent knowledge but struggles to retain information from earlier domains. Moreover, the most recent segment $\mathcal{A}^T_{16:20}$ also degrades at IPC = 20, suggesting that $\mathcal{L}_{\text{stable}}$ can hinder the plasticity of the synthetic dataset. In contrast, applying the asymmetric update strategy via bi-level optimization improves performance across all segments, including $\mathcal{A}^T_{1:5}$ and $\mathcal{A}^T_{16:20}$, indicating that our method effectively balances stability and plasticity.

Moreover, to verify the effectiveness of the proposed bi-level optimization method, we implement a simple alternative that assigns per sample $\bar{\alpha}_i$ and $\bar{\beta}_i$. The most naive approach is to assign linearly decreasing and increasing values to $\bar{\alpha}_i$ and $\bar{\beta}_i$ for the sample indexed by $i$ in the synthetic dataset, respectively. Specifically, we define $\bar{\alpha}_i = \alpha_{\max} - \frac{i}{N-1}\left(\alpha_{\max} - \alpha_{\min}\right)$ and $\bar{\beta}_i = \beta_{\min} + \frac{i}{N-1}\left(\beta_{\max} - \beta_{\min}\right)$, where $i \in \{0, \ldots, N-1\}$. The results are shown in the third row of Table 3, and we observe that this linear assignment fails to consistently improve performance across all segments, implying that our bi-level optimization method more effectively assigns $\bar{\alpha}_i$ and $\bar{\beta}_i$ than the linear assignment scheme.

| IPC | 10 | | | | 20 | | | |
|---|---|---|---|---|---|---|---|---|
| Method | $\mathcal{A}^T_{1:5}$ | $\mathcal{A}^T_{6:10}$ | $\mathcal{A}^T_{11:15}$ | $\mathcal{A}^T_{16:20}$ | $\mathcal{A}^T_{1:5}$ | $\mathcal{A}^T_{6:10}$ | $\mathcal{A}^T_{11:15}$ | $\mathcal{A}^T_{16:20}$ |
| Finetune | $33.4_{\pm0.7}$ | $19.4_{\pm1.1}$ | $23.7_{\pm2.6}$ | $80.9_{\pm6.6}$ | $31.3_{\pm2.0}$ | $19.7_{\pm2.6}$ | $31.0_{\pm4.8}$ | $85.6_{\pm3.4}$ |
| + $\mathcal{L}_{\text{stable}}$ | $26.9_{\pm0.7}$ | $27.5_{\pm2.2}$ | $59.3_{\pm2.9}$ | $87.1_{\pm1.9}$ | $29.4_{\pm3.7}$ | $25.5_{\pm2.7}$ | $45.0_{\pm3.5}$ | $77.5_{\pm4.5}$ |
| + Asym. (Linear) | $31.0_{\pm1.9}$ | $33.4_{\pm3.9}$ | $56.6_{\pm8.3}$ | $82.7_{\pm0.6}$ | $27.3_{\pm3.1}$ | $27.3_{\pm1.9}$ | $48.7_{\pm3.2}$ | $81.1_{\pm2.6}$ |
| + Asym. (Bi-level) | $\mathbf{36.3_{\pm1.1}}$ | $\mathbf{43.8_{\pm4.7}}$ | $\mathbf{76.3_{\pm3.3}}$ | $\mathbf{94.5_{\pm0.3}}$ | $\mathbf{32.9_{\pm1.0}}$ | $\mathbf{34.8_{\pm1.8}}$ | $\mathbf{72.9_{\pm0.7}}$ | $\mathbf{95.2_{\pm0.1}}$ |

Table 3: Ablation study on R-MNIST when IPC = 10 and IPC = 20. Asym. denotes the proposed asymmetric update method, with two strategies: linear assignment and the proposed bi-level optimization.

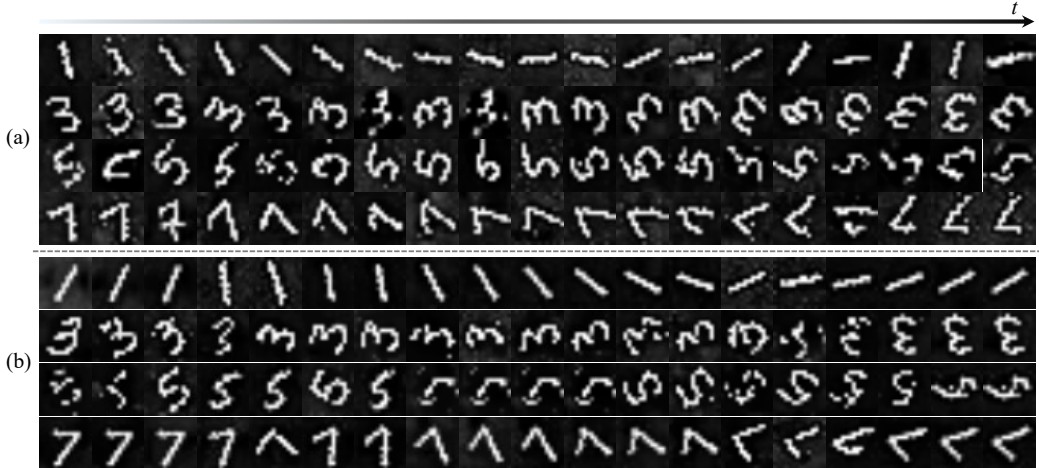

Figure 6: Synthesized images on R-MNIST for classes 1, 3, 5, and 7 across $t$. (a) and (b) show images with the smallest and largest values of $\bar{\alpha}_i - \bar{\beta}_i$ for each class at each $t$, respectively.

We also visualize synthesized images prioritized for stability and plasticity, respectively. Specifically, samples with the smallest values of $\bar{\alpha}_i - \bar{\beta}_i$ are regarded as plasticity focused, whereas those with the largest values are stability focused. Figure 6 shows results on R-MNIST for classes 1, 3, 5, and 7 across domains. (a) displays plasticity focused samples, which mainly capture knowledge from the most recent domain and exhibit the domain specific rotation angles. By contrast, stability focused samples in (b) appear more consistent across domains, indicating preservation of earlier knowledge. These observations support that our bi-level optimization method assigns $\bar{\alpha}_i$ and $\bar{\beta}_i$ effectively to balance stability and plasticity.

## 6 CONCLUSION

In this work, we introduced a novel problem setting, *Domain Incremental Dataset Distillation* (DIDD), which aims to continually distill a sequence of datasets from different domains into a single synthetic dataset. By maintaining a fixed-size synthetic dataset across time, DIDD keeps both storage and training costs efficient regardless of the number of datasets. Since exisiting DD methods only focus on the plasticity of the synthetic dataset, they suffer from catastrophic forgetting. To address this problem, we introduced stability loss that preserves knowledge from earlier datasets. Moreover, to balance between stability and plasticity, we proposed an asymmetric update strategy that assigns different update rates for plasticity and stability to each sample. To effectively determine those update rates, we presented a bi-level optimization method based on meta-learning. It encourages each sample to be specialized in either stability or plasticity, effectively reducing their inherent conflict. Extensive experiments validated the effectiveness of our method, showing consistent improvements over existing continual learning methods adapted to the DIDD task.

## ACKNOWLEDGEMENTS

This work was supported in part by the National Research Foundation of Korea (NRF) grant funded by the [Ministry of Science and ICT (MSIT)] under Grant RS-2024-00392536, in part by the Institute of Information and Communications Technology Planning and Evaluation (IITP) grants funded by the Korean Government (MSIT), including the Leading Generative Artificial Intelligence (AI) Human Resources Development Program under Grant IITP-2025-RS-2024-00360227, in part by the Artificial Intelligence Graduate School Program [Ulsan National Institute of Science and Technology (UNIST)] under Grant RS-2020-II201336, and in part by the AI Star Fellowship Program (UNIST) under Grant RS-2025-25442824.

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

# A  APPENDIX

Large language models (LLMs) were used solely for language refinement, and all research content was generated entirely by the authors.

## A.1  ALGORITHM

Algorithm 1 summarizes the overall procedure of our proposed method.

## A.2  SPECIFICATION OF DATASETS

We summarize the specifications of each dataset in Table 4. We also visualize examples of R-MNIST, Seq-CORe50, and PACS in Figure 7, 8 and 9. Each column corresponds to a different domain and each row corresponds to a different class.

| Dataset | R-MNIST | Seq-CORe50 | PACS |
|---|---|---|---|
| # of classes | 10 | 50 | 7 |
| # of domains | 20 | 11 | 4 |
| Average # of images per domain (train) | 60,000 | 11,990 | 2,244 |
| Average # of images per domain (test) | 10,000 | 2,999 | 715 |
| Total # of images (train) | 1,200,000 | 131,888 | 8,977 |
| Total # of images (test) | 200,000 | 32,987 | 1,014 |

Table 4: Specification of datasets.

---

**Algorithm 1** Asymmetric Synthetic Data Update for DIDD

---

**Require:** Datasets $\{\mathcal{D}^t\}_{t=1}^T$ with label set $\mathcal{C}$; IPC $N$; randomly initialized model $\phi$; total iterations $I$; learning rates $\eta_x, \eta_\alpha, \eta_\beta$; $\alpha, \beta$ bounds $(\alpha_{\min}, \alpha_{\max}), (\beta_{\min}, \beta_{\max})$; hyperparameters $\lambda_\alpha, \lambda_\beta$.

1: **for** $t = 1$ to $T$ **do**
2:      $\hat{\mathcal{D}}^t \leftarrow \hat{\mathcal{D}}^{t-1}$; **freeze** $\hat{\mathcal{D}}^{t-1}$
3:      **for** iter $= 1$ to $I$ **do**
4:          **for** each class $c \in \mathcal{C}$ **do**
5:              Sample a class $c$ mini-batch from $\mathcal{D}_t$
6:              Compute $F(\boldsymbol{x}^t), F(\hat{\boldsymbol{x}}^t), F(\hat{\boldsymbol{x}}^{t-1})$ by $\phi$.
7:              Compute $\mathcal{L}_{\mathrm{plastic}}, \mathcal{L}_{\mathrm{stable}}$ using Eqs. 3 and 4
8:              Get gradients $\boldsymbol{g}_{\mathcal{P},i}, \boldsymbol{g}_{\mathcal{S},i}$ via Eq. 6.
9:              **Meta update:** $\hat{\boldsymbol{x}}_{\mathrm{meta},i}^t$ by Eq. 9.
10:            Compute meta-losses $\mathcal{L}_{\mathrm{meta}}$ (Eq. 10) and $\mathcal{L}_{\mathrm{penalty}-\alpha}, \mathcal{L}_{\mathrm{penalty}-\beta}$ (Eq. 12).
11:            **Update scale parameters:** $\alpha_i, \beta_i$ by Eq. 14.
12:            **Asymmetric update:** $\hat{\boldsymbol{x}}_i^t$ by Eq. 7.
13:          **end for**
14:      **end for**
15: **end for**
16: **return** $\hat{\mathcal{D}}^T$.

---

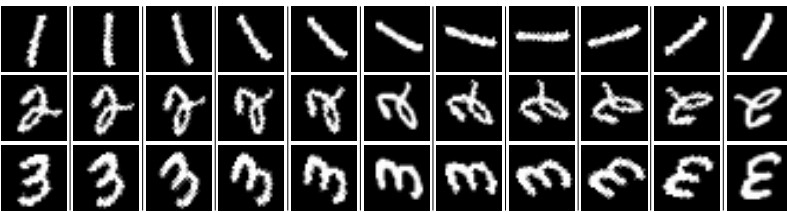

Figure 7: Examples for each domain in R-MNIST.

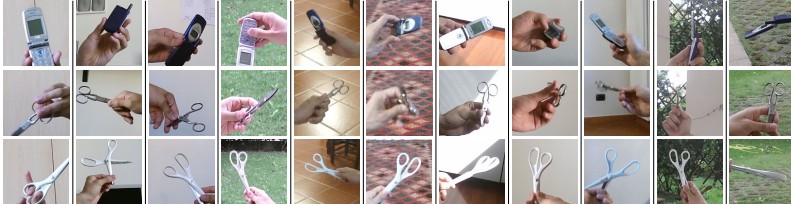

Figure 8: Examples for each domain in Seq-CORe50.

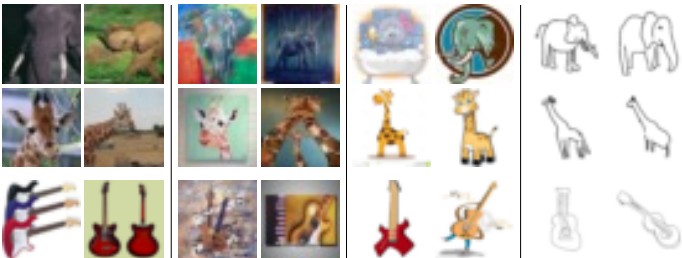

Figure 9: Examples for each domain in PACS.

## A.3 COMPARISON OF STORAGE COST

We compare storage over time on R-MNIST, Seq-CORe50, and PACS with IPC $= 20$, assuming a new dataset arrives at each time step $t$. Figure 10 plots time $t$ on the $x$-axis and storage in MB on a log scale on the $y$-axis. As $t$ increases, the size required to store the original datasets grows rapidly.

Conventional DD greatly reduces storage, yet its efficiency diminishes over time because distilled sets accumulate for each domain. In contrast, our method maintains a fixed size synthetic dataset across time, so the storage remains constant regardless of $t$.

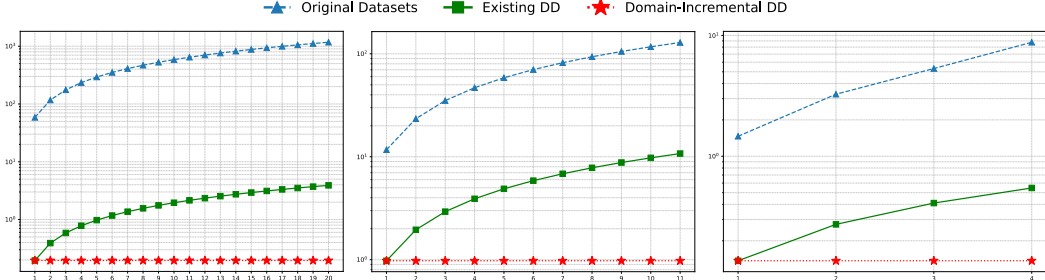

Figure 10: Comparison of storage cost on R-MNIST (left), Seq-CORe50 (middle), and PACS (right) when IPC = 20. $x$-axis indicates $t$ and $y$-axis indicates the storage cost in MB (log scale).

## A.4 EFFECT OF DIFFERENT SEQEUNCE ORDERS

We conducted additional experiments with different orders of training sequences. Table 5 reports the results with randomly generated two different orders (2, 3), in addition to the default order (1), for R-MNIST and PACS datasets. We observe that the proposed method consistently outperforms the baselines with different sequence orders, demonstrating its robustness to the sequence orders.

| Dataset | R-MNIST | | | | | | PACS | | | | | |
|---|---|---|---|---|---|---|---|---|---|---|---|---|
| Order | 1 | | 2 | | 3 | | 1 | | 2 | | 2 | |
| Metric | $\mathcal{A}^T$ (↑) | $\mathcal{F}^T$ (↓) | $\mathcal{A}^T$ (↑) | $\mathcal{F}^T$ (↓) | $\mathcal{A}^T$ (↑) | $\mathcal{F}^T$ (↓) | $\mathcal{A}^T$ (↑) | $\mathcal{F}^T$ (↓) | $\mathcal{A}^T$ (↑) | $\mathcal{F}^T$ (↓) | $\mathcal{A}^T$ (↑) | $\mathcal{F}^T$ (↓) |
| Finetune | 39.4±2.2 | 56.7±2.0 | 42.7±1.7 | 53.6±1.6 | 40.5±0.3 | 56.0±0.4 | 26.9±0.8 | 33.2±0.3 | 35.4±0.9 | 24.0±0.6 | 25.7±2.0 | 32.4±2.0 |
| MAS | 45.6±0.9 | 49.1±0.8 | 46.9±1.0 | 47.7±0.9 | 44.2±0.8 | 50.5±0.8 | 35.0±1.7 | 6.0±1.0 | 37.7±1.0 | 4.9±1.3 | 40.6±2.9 | 7.2±3.4 |
| Proposed | 62.7±2.0 | 33.1±2.0 | 62.5±2.0 | 33.3±2.0 | 63.7±2.4 | 32.1±2.4 | 48.0±3.1 | 8.4±1.8 | 48.0±1.2 | 8.2±1.1 | 50.7±0.7 | 7.1±1.4 |

Table 5: Performance comparison on R-MNIST and PACS datasets under IPC=10 with different 3 dataset orders.

## A.5 HYPERPARAMETERS

For simplicity, most hyperparameters are shared across datasets and IPC settings. We summarize them in Table 6. Only for PACS, we set the batch size to 64 because there exists a class with fewer than 128 images. The image update learning rate $\eta_x$ is set equal to the IPC. For example, $\eta_x = 1$ when IPC = 1 and $\eta_x = 10$ when IPC = 10, following (Zhang et al., 2024).

| Iteration | Batch size | $\eta_x$ | $\eta_\alpha$ | $\eta_\beta$ | $\lambda_\alpha$ | $\lambda_\beta$ | $\alpha_{min}$ | $\alpha_{max}$ | $\beta_{min}$ | $\beta_{max}$ |
|---|---|---|---|---|---|---|---|---|---|---|
| 3,000 | 128* | same as IPC | 1e-2 | 1e-2 | 1e-4 | 1e-4 | 0 | 2 | 0 | 2 |

* For PACS only, the batch size is set to 64.

Table 6: Hyperparameter settings.

### A.5.1 ANALYSIS ON $\lambda_\alpha$ AND $\lambda_\beta$

We analyze the impact of $\lambda_\alpha$ and $\lambda_\beta$. Table 7 reports results on R-MNIST with IPC = 10. Increasing $\lambda_\beta$ with $\lambda_\alpha$ improves performance on earlier segments, because a stronger penalty on $\beta_i$ reduces updates for plasticity. As a result, the synthetic dataset more focuses on stability. Conversely, increasing $\lambda_\alpha$ with fixed $\lambda_\beta$ improves performance on recent segments, since update rates for stability shrinks while increasing plasticity relatively. These results support that our bi-level optimization method works as intended to balance between stability and plasticity by adjusting $\lambda_\alpha$ and $\lambda_\beta$.

| IPC | | 10 | | | | IPC | | 10 | | | |
|---|---|---|---|---|---|---|---|---|---|---|---|
| $\lambda_\alpha$ | $\lambda_\beta$ | $\mathcal{A}^T_{1:5}$ | $\mathcal{A}^T_{6:10}$ | $\mathcal{A}^T_{11:15}$ | $\mathcal{A}^T_{16:20}$ | $\lambda_\alpha$ | $\lambda_\beta$ | $\mathcal{A}^T_{1:5}$ | $\mathcal{A}^T_{6:10}$ | $\mathcal{A}^T_{11:15}$ | $\mathcal{A}^T_{16:20}$ |
| | 5e-5 | 36.0 | 44.5 | 75.6 | **95.1** | | 5e-5 | **39.9** | 42.0 | 76.2 | 94.1 |
| 1e-4 | 1e-4 | 36.3 | 43.8 | 76.3 | 94.5 | 1e-4 | 1e-4 | 36.3 | **43.8** | 76.3 | 94.5 |
| | 5e-4 | **37.5** | **48.8** | **79.0** | 93.9 | | 5e-4 | 35.1 | 41.5 | **78.9** | **95.2** |

Table 7: Perfomance on R-MNIST when IPC = 10 to study the impact of $\lambda_\alpha$ and $\lambda_\beta$.

## A.6 SCAILABILITY TO LARGER BACKBONES AND DATASET

Table 8 presents the performance of the proposed method when scaling up to larger back-bone (ResNet) and higher image resolution (128×128). Seq-CORe50 dataset can be regarded as medium scale one, which is originally in 128×128 resolution and has 50 classes. The results show that the proposed method outperforms MAS method, which is the second-best baseline on most cases, with clear margins, thereby demonstrating its effectiveness and scalability to larger back-bones and datasets.

| Resolution | 32×32 | | 128×128 | |
|---|---|---|---|---|
| Backbone | ConvNet | | ResNet | |
| IPC | 10 | | 10 | |
| Metric | $\mathcal{A}^T$ (↑) | $\mathcal{F}^T$ (↓) | $\mathcal{A}^T$ (↑) | $\mathcal{F}^T$ (↓) |
| Finetune | 25.4 | 79.9 | 26.5 | 65.3 |
| MAS | 26.4 | 60.3 | 26.5 | 11.7 |
| Proposed | 55.4 | 38.8 | 61.7 | 28.2 |

Table 8: Performance on Seq-CORe50 under IPC=10 with different backbone and image resolution.

We additionally evaluated the performance of the proposed method on six ImageNet-based subsets: ImageNette, ImageWoof, ImageMeow, ImageFruit, ImageYellow, and ImageSquawk, under IPC=1. Since ImageNet does not provide explicit domains, we constructed five domain sequences by se-lecting five corruption types from ImageNet-C Hendrycks & Dietterich (2019) (defocus blur, fog, speckle noise, contrast, and frost). We tested two high-capacity backbone models, ResNet and ViT. The results are summarized in Table 9. Although ViT has larger model capacity, its dataset dis-tillation performance is generally lower than that of ResNet, which aligns with the observation in NRR-DD Tran et al. (2025) that stronger backbones do not necessarily yield better distillation per-formance. Nevertheless, the proposed method outperforms the Finetune baseline in most cases and both backbone models, demonstrating that our framework generalizes well to large-scale datasets and modern architectures.

| Dataset | ImageNette | | | | ImageWoof | | | | ImageMeow | | | |
|---|---|---|---|---|---|---|---|---|---|---|---|---|
| Backbone | ResNet | | ViT | | ResNet | | ViT | | ResNet | | ViT | |
| IPC | 1 | | 1 | | 1 | | 1 | | 1 | | 1 | |
| Metric | $\mathcal{A}^T$ (↑) | $\mathcal{F}^T$ (↓) | $\mathcal{A}^T$ (↑) | $\mathcal{F}^T$ (↓) | $\mathcal{A}^T$ (↑) | $\mathcal{F}^T$ (↓) | $\mathcal{A}^T$ (↑) | $\mathcal{F}^T$ (↓) | $\mathcal{A}^T$ (↑) | $\mathcal{F}^T$ (↓) | $\mathcal{A}^T$ (↑) | $\mathcal{F}^T$ (↓) |
| Joint | 40.6 | - | 18.6 | - | 24.2 | - | 14.4 | - | 25.2 | - | 15.6 | - |
| Finetune | 32.4 | 10.0 | 18.0 | 9.8 | 21.6 | 1.3 | 12.4 | 6.3 | 21.0 | 9.3 | 14.8 | 8.5 |
| Proposed | 42.8 | 1.3 | 19.5 | 4.8 | 23.0 | 0.5 | 15.8 | 2.3 | 18.2 | 6.3 | 15.2 | 6.8 |

| Dataset | ImageFruit | | | | ImageYellow | | | | ImageSquawk | | | |
|---|---|---|---|---|---|---|---|---|---|---|---|---|
| Backbone | ResNet | | ViT | | ResNet | | ViT | | ResNet | | ViT | |
| IPC | 1 | | 1 | | 1 | | 1 | | 1 | | 1 | |
| Metric | $\mathcal{A}^T$ (↑) | $\mathcal{F}^T$ (↓) | $\mathcal{A}^T$ (↑) | $\mathcal{F}^T$ (↓) | $\mathcal{A}^T$ (↑) | $\mathcal{F}^T$ (↓) | $\mathcal{A}^T$ (↑) | $\mathcal{F}^T$ (↓) | $\mathcal{A}^T$ (↑) | $\mathcal{F}^T$ (↓) | $\mathcal{A}^T$ (↑) | $\mathcal{F}^T$ (↓) |
| Joint | 27.2 | - | 27.8 | - | 50.6 | - | 30.4 | - | 40.4 | - | 26.6. | - |
| Finetune | 25.0 | 4.3 | 20.2 | 7.8 | 43.0 | 4.3 | 20.6 | 14.5 | 36.6 | 6.0 | 25.2 | 2.3 |
| Proposed | 31.2 | 1.8 | 22.2 | 3.5 | 43.4 | 0.0 | 28.2 | 8.0 | 42.8 | 0.5 | 27.6 | 1.3 |

Table 9: Performance on ImageNet-C subsets, under IPC=1 with two different backbones.

## A.7 OPTIMIZATION DIFFICULTY IN META-LEARNING FRAMEWORK

Conventional meta-learning frameworks often encounter optimization difficulties arising from the nested structure of their objectives, where large number of inner loop updates may lead to vanishing

or exploding gradients. Let $w_0$ be the initial parameters that we aim to optimize, $\mathcal{L}_{\mathrm{tr}}$ be the training loss for inner loop updates, and $\mathcal{L}_{\mathrm{meta}}$ be the meta-loss for outer loop updates. In the inner loop, the parameters are updated as:

$$w_t = w_{t-1} - \eta \nabla_{w_{t-1}} \mathcal{L}_{\mathrm{tr}}(w_{t-1}), \quad t = 1, 2, \ldots, T, \tag{20}$$

where $\eta$ is the inner loop learning rate and $T$ is the number of inner loop updates. In the outer loop, $w_0$ is updated based on the meta-loss with respect to $w_T$:

$$w_0 \leftarrow w_0 - \beta \nabla_{w_0} \mathcal{L}_{\mathrm{meta}}(w_T), \tag{21}$$

where $\beta$ is the outer loop learning rate. Here, the gradient $\nabla_{w_0} \mathcal{L}_{\mathrm{meta}}(w_T)$ can be expressed using the chain rule as:

$$\nabla_{w_0} \mathcal{L}_{\mathrm{meta}}(w_T) = \nabla_{w_T} \mathcal{L}_{\mathrm{meta}}(w_T) \cdot \prod_{t=1}^{T} \left( I - \eta \nabla_{w_{t-1}}^2 \mathcal{L}_{\mathrm{tr}}(w_{t-1}) \right). \tag{22}$$

We can observe that the gradient can either vanish or explode with increasing $T$. However, the proposed method employs only a single inner loop update ($T = 1$), and thus effectively mitigates these optimization challenges, leading to more stable and efficient training.

## A.8 LIMITATIONS AND FUTURE WORK

In this work, we focus on the domain incremental setting, where datasets from different domains arrive sequentially and share the same label space. However, our approach does not yet address other continual learning settings, such as class incremental learning where new classes are introduced over time. Extending the DIDD framework to handle the arrival of new classes is an important direction for future work. We will explore strategies to expand the synthetic dataset to incorporate new classes while preserving knowledge of existing ones, aiming for a more versatile and widely applicable dataset distillation framework.

## A.9 MORE VISUALIZATION

We visualize more synthesized images on R-MNIST, Seq-CORe50, and PACS comparing Finetune and our proposed method in Figures 11, 12, and 13, respectively.

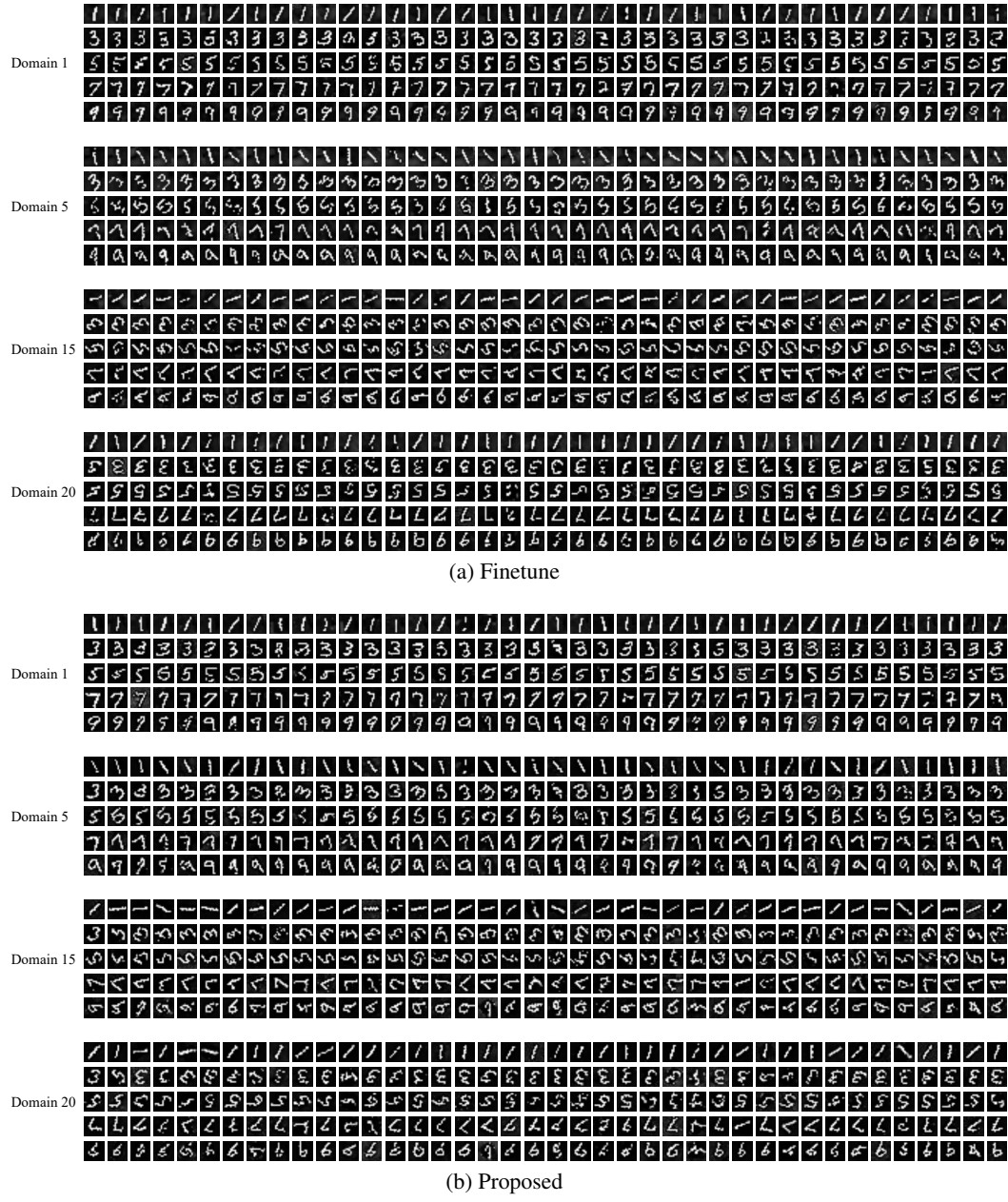

Figure 11: Synthesized images on R-MNIST for (a) Finetune and (b) the proposed method.

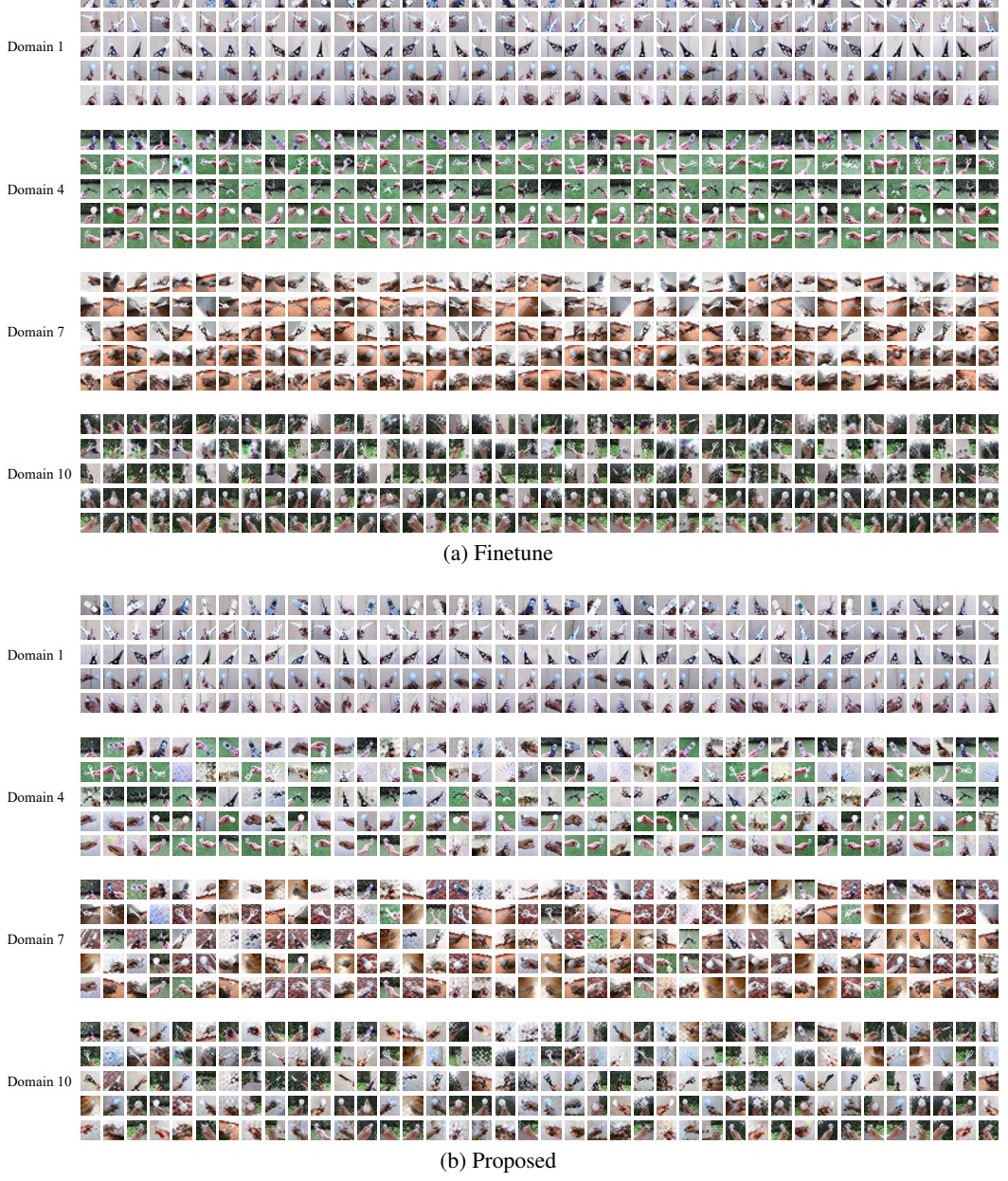

Figure 12: Synthesized images on Seq-CORe50 for (a) Finetune and (b) the proposed method.

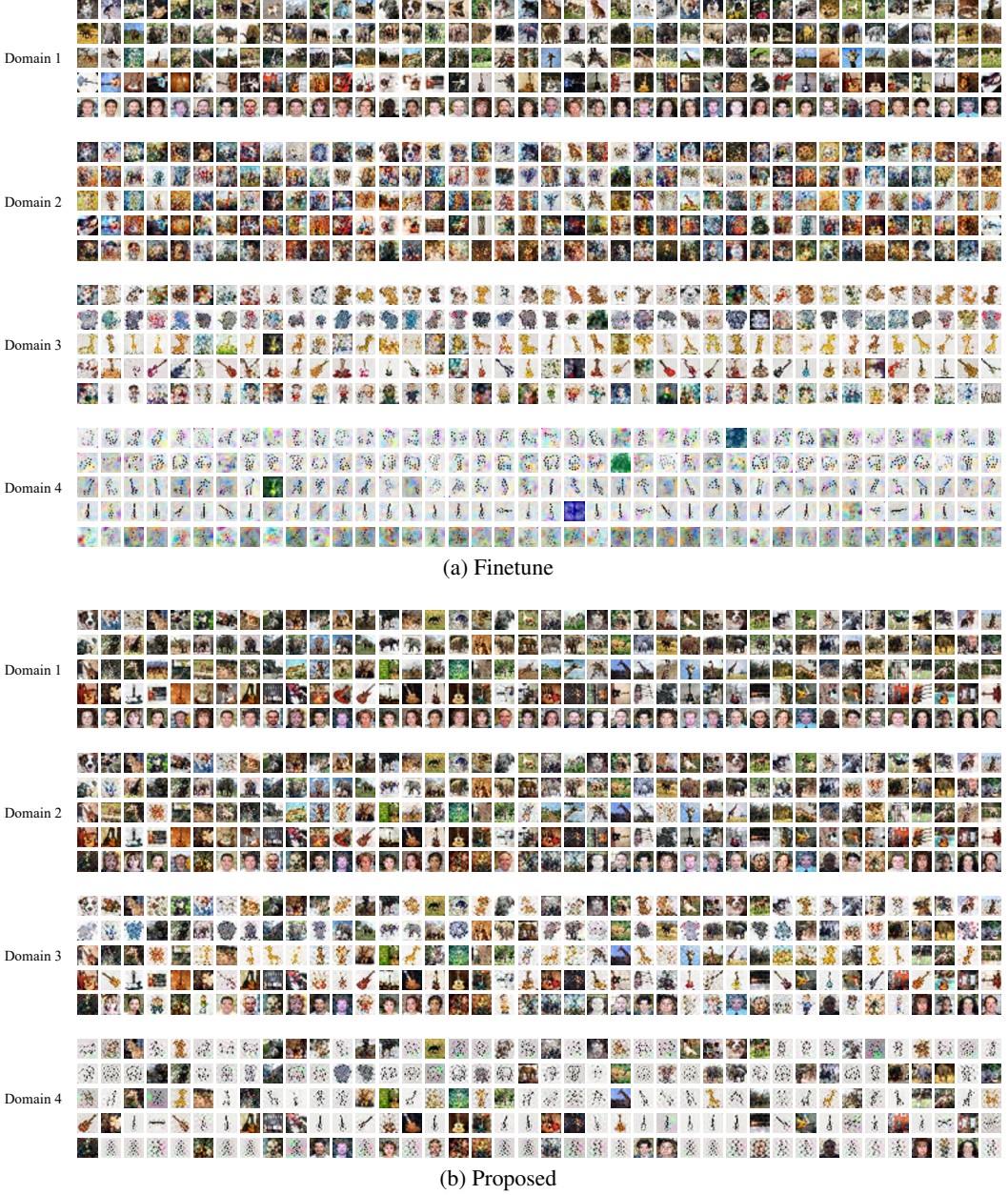

Figure 13: Synthesized images on PACS for (a) Finetune and (b) the proposed method.

