# OpenReview forum: "Asymmetric Synthetic Data Update for Domain Incremental Dataset Distillation"
_ICLR.cc/2026/Conference — ICLR 2026 Poster_

### Official Review · Reviewer_7mfs · 2025-10-15

**Soundness:** 3
**Presentation:** 3
**Contribution:** 3
**Rating:** 6
**Confidence:** 3

**Summary:**

This paper introduces Domain Incremental Dataset Distillation (DIDD) — a new problem setting where synthetic data must be updated continuously as new domains arrive, under a fixed memory budget.
Unlike conventional dataset distillation, which assumes access to all data at once, DIDD requires the synthetic set to maintain past knowledge while integrating new information, analogous to continual learning but at the *data level* rather than *parameter level*.

To tackle this, the authors propose Asymmetric Synthetic Data Update, a bi-level optimization framework where each synthetic sample learns its own *stability–plasticity* trade-off coefficients. These coefficients adaptively balance gradients from old and new domain objectives, allowing the synthetic dataset to evolve without catastrophic forgetting.

Experiments on R-MNIST, Seq-CORe50, and PACS demonstrate significant gains and reduced forgetting over standard distillation and continual learning baselines. Ablation analyses show the importance of asymmetric updates and bi-level learning for maintaining cross-domain knowledge.

**Strengths:**

* **Novel problem definition:** The paper bridges the gap between dataset distillation and continual learning, providing a fresh research direction.
* **Conceptually elegant solution:** The asymmetric update mechanism offers an intuitive and interpretable way to balance stability and plasticity at the data level. The per-sample α/β weighting is original and theoretically motivated.
* **Strong empirical gains on small-scale benchmarks:** The proposed method outperforms both conventional dataset distillation methods (e.g., MTT, DSA) and continual learning baselines (e.g., EWC, LwF, MAS), validating the effectiveness of the asymmetric strategy.

**Weaknesses:**

* **Scalability concerns:** The method relies on *bi-level optimization for each sample*, which is computationally expensive. The current experiments use small networks (3-layer ConvNet) and small datasets (R-MNIST, PACS). It is unclear whether the approach can scale to modern architectures (ResNet, ViT) or larger datasets (CIFAR-100, ImageNet).
* **Lack of large-scale validation:** The paper would be significantly stronger if it included experiments or runtime analysis on medium-scale settings to demonstrate practical feasibility.
* **Limited diversity of tasks:** The benchmarks are all domain-incremental vision tasks; it would be interesting to see if this framework can handle class-incremental or multimodal distillation scenarios.
* **Ablation on efficiency missing:** Although the method is theoretically well-motivated, there is no quantification of time or memory overhead compared to one-shot distillation.

**Questions:**

1. How does training time scale with dataset size or number of domains? Can the authors provide runtime or complexity analysis?
2. Could the asymmetric update be approximated with parameter-efficient strategies (e.g., low-rank updates, meta-network sharing) to improve scalability?
3. How would the approach behave on *class-incremental* or *multi-modal* tasks rather than domain-incremental ones?
4. Is it possible to replace full bi-level optimization with a single-level approximation or gradient truncation without significant performance loss?
5. Could this framework generalize to larger backbones (e.g., ResNet, ViT) or to non-vision modalities (text or graph distillation)?

---

> ### Author Response · Authors · 2025-11-20
> **Response for Reviewer 7mfs (1/2)**
>
> ### **A1. Scailability to Large Backbone and Dataset**
> Seq-CORe50, used in our experiments, is regarded as a medium scale dataset, which has 50 classes and images with 128$\times$128 resolution. In this work, we downsampled the images in Seq-CORe50 into 32$\times$32 resolution. We additionally evaluated the performance of the proposed method when using a larger backbone (ResNet) and a higher image resolution (128$\times$128) in the table below.
> The results show that while the MAS method, the second-best baseline on most cases, fails to scale up with larger backbone and image resolutions, the proposed method successes to scale up and achieves better performance with clear margin, thereby demonstrating its effectiveness and scalability.
> We included this in the revised manuscript. Please see Appendix A.5.
> |Seq-CORe50|32$\times$32|ConvNet|128$\times$128|ResNet|
> |--------|---------|---------|---------|---------|
> |Metric| 𝓐ᵀ (↑) | 𝓕ᵀ (↓) | 𝓐ᵀ (↑) | 𝓕ᵀ (↓)
> | Finetune | 25.4 | 79.9 | 26.5 | 65.3 |
> | MAS | 26.4 | 60.3 | 26.5 | 11.7 |
> |Proposed | 55.4|38.8|61.7|28.2
>
> ### **A2. Extending to Class-Incremental or Multimodal Scenarios**
> Thanks for the insightful comment. Class Incremental Dataset Distillation (CIDD) or multimodal distillation are additional new problems that are beyond the scope of this paper.
> As we discussed in Appendix, our future research direction includes the application of our methodology to CIDD or multimodal scenarios.
>
> ### **A3. Complexity Analysis**
> As the reviewer suggested, we provided a detailed complexity analysis on the proposed method evaluated on the PACS dataset when IPC=10. As shown in the table below, the proposed method incurs a higher distillation cost compared to other baselines due to the additional process of bi-level optimization.
> However, it is important to note that DD is fundamentally an **OFFLINE** procedure executed only **ONCE** prior to model training, and thus does not incur the complexity associated with real-time processing.
> In addition, we conducted an experiment to scale the distillation cost by adjusting the number of iterations as 1,000 during the distillation process.
> Note that, even with a smaller number of iterations, the proposed method (Proposed~(1,000 iter)) still outperforms other baselines significantly, while achieving the lowest distillation cost. We included this result in Section 5.2 (Line 448)  of the revised manuscript.
> | Method                | **P (s)** | **A (s)** | **C (s)** | **S (s)** | **Total (s)** | **𝓐ᵀ (↑)**        | **𝓕ᵀ (↓)**        |
> |-----------------------|-----------|-----------|-----------|-----------|---------------|---------------------|---------------------|
> | **Finetune**          | 112.2     | 102.1     | 101.9     | 94.6      | 410.7         | 26.9 $\pm$ 0.8      | 44.3 $\pm$ 0.3      |
> | **LwF**               | 109.4     | 145.2     | 130.4     | 139.0     | 524.0         | 35.0 $\pm$ 0.6      | 13.0 $\pm$ 0.3      |
> | **EWC**               | 112.5     | 121.9     | 109.8     | 114.1     | 458.3         | 26.4 $\pm$ 0.7      | 43.7 $\pm$ 0.5      |
> | **LF**                | 103.8     | 151.0     | 143.9     | 141.4     | 540.0         | 27.4 $\pm$ 3.4      | 42.9 $\pm$ 2.4      |
> | **MAS**               | 107.5     | 117.3     | 103.6     | 97.5      | 425.8         | 35.0 $\pm$ 1.7      | 7.9 $\pm$ 1.0       |
> | **Proposed**          | 111.4     | 342.5     | 334.2     | 335.1     | 1123.3        | 48.0 $\pm$ 3.1      | 11.2 $\pm$ 1.8      |
> | **Proposed (1000 iter)** | 37.5  | 109.6     | 99.6      | 97.4      | 344.1         | 46.3 $\pm$ 1.8      | 8.5 $\pm$ 1.2       |
>
> Furthermore, to analyze the complexity of the proposed method with respect to the image resolution, we conducted experiments on the PACS dataset with varying image resolutions: 32$\times$32, 64$\times$64, and 128$\times$128.
> Table below summarizes the distillation time and peak memory usage according to different image resolutions.
> We observe that the distillation cost linearly increases with the number of domains and the image resolution.
> Specifically, doubling the resolution results in approximately a four-fold increase in both distillation time and peak memory usage.
> Notably, the proposed method only requires around 3.8GB of the peak memory even at the high resolution of 128$\times$128, demonstrating its scalability to larger datasets.
> | Method                | **P (s)** | **A (s)** | **C (s)** | **S (s)** | **Total (s)** | **Peak Memory (MB)**        |
> |-----------------------|-----------|-----------|-----------|-----------|---------------|---------------------|
> | 32$\times$32       |111.4|342.5|334.2|335.1|1,123.3|270.8|
> | 64$\times$64      | 175.4|499.2|492.9|490.8|1,658.2|987.2|
> | 128$\times$128       | 504.8|1,562.7|1,561.4|1,547.0|5,175.9|3,847.2|

---

> ### Author Response · Authors · 2025-11-20
> **Response for Reviewer 7mfs (2/2)**
>
> ### **A4. Parameter-efficient Update**
> We appreciate the reviewer's valuable suggestion.
> However, we would like to highlight that the number of required parameters of $\boldsymbol{\alpha}$ and $\boldsymbol{\beta}$ is only $2\times N$, where $N$ is the number of synthetic samples, which is constant across entire domains and negligible compared to the number of model parameters.
>
> ### **A5. Single-Level Approximation of Meta-Learning**
> As the reviewer suggested, we conducted experiments using the single-level approximation.
>
> Let $\hat{x}$ be the synthetic data and ${\hat{x}}\_\mathrm{meta}$ be the meta data obtained after the inner-loop update.
> $$
> {\hat{x}}\_\mathrm{meta}=\Phi(\hat{x})=\hat{x}-\eta_x\left(\sum_{i=1}^{N}\bar{\alpha}\_i \cdot \boldsymbol{g}_{\mathcal{S},i}+\bar{\beta}_i \cdot {\boldsymbol{g}}\_{\mathcal{P},i}\right)=\hat{x}-\eta\_x \, \mathbf{G}(\hat{x}, \alpha, \beta).
> $$
>
> The meta loss is formulated as:
> $$\mathcal{L}\_{\mathrm{meta}}(\hat{x})=\tilde{\mathcal{L}}(\hat{x})+\lambda_{\alpha}\mathcal{L}\_{\mathrm{penalty}-\alpha}+\lambda_{\beta}\mathcal{L}_{\mathrm{penalty}-\beta}.
> $$
>
> where
> $$\tilde{\mathcal{L}}(\hat{x})=\mathcal{L}\_{\mathrm{stable}}(\hat{x})+\mathcal{L}_{\mathrm{plastic}}(\hat{x}).
> $$
>
> We approximate $\tilde{\mathcal{L}}(\hat{x})$ using the first-order Taylor expansion:
> $$
> \tilde{\mathcal{L}}(\hat{x}) \approx \tilde{\mathcal{L}}(\hat{x}) + \nabla\_{\hat{x}} \tilde{\mathcal{L}}(\hat{x})^\top
> (\hat{x} - \hat{x}) = \tilde{\mathcal{L}}(\hat{x}) - \eta_x \langle \nabla\_{\hat{x}} \tilde{\mathcal{L}}(\hat{x}), \mathbf{G}(\hat{x}, \alpha, \beta) \rangle.
> $$
>
> Consequently, the approximate meta loss becomes:
> $$
> \mathcal{L}\_{\mathrm{meta}}(\hat{x})\approx\tilde{\mathcal{L}}(\hat{x})-\eta_x\langle\nabla_{\hat{x}} \tilde{\mathcal{L}} (\hat{x}), \mathbf{G}(\hat{x}, \alpha, \beta) \rangle + \lambda_{\alpha}\mathcal{L}\_{\mathrm{penalty}-\alpha} +
> \lambda_{\beta}\mathcal{L}\_{\mathrm{penalty}-\beta}.
> $$
>
> This approximate objective is minimized with respect to $\hat{x}$, $\boldsymbol{\alpha}$, and $\boldsymbol{\beta}$.
> The experimental results on the PACS dataset under IPC=10 are summarized in the table below (Approx.).
>
> Unfortunately, the single-level approximation degrades performance by a noticeable margin, performing worse than the variant that removes the asymmetric update strategy and uses only the stability loss.
> | Baseline            |                     | + L_stable        |                     | Approx.           |                     | Proposed          |                     |
> |---------------------|---------------------|--------------------|----------------------|--------------------|----------------------|--------------------|----------------------|
> | $\mathcal{A}^T$ (↑) | $\mathcal{F}^T$ (↓) | $\mathcal{A}^T$ (↑) | $\mathcal{F}^T$ (↓) | $\mathcal{A}^T$ (↑) | $\mathcal{F}^T$ (↓) | $\mathcal{A}^T$ (↑) | $\mathcal{F}^T$ (↓) |
> | 26.9                | 44.3                | 44.7               | 10.9                 | 42.8               | 9.4                  | 48.0               | 8.4                  |

---

> ### Comment · Reviewer_7mfs · 2025-11-21
> **Thank you for the detailed rebuttal and additional experimental results.**
>
> Most of my concerns were addressed satisfactorily.
>
> However, I still have reservations regarding scalability to truly large-scale benchmarks and modern architectures (e.g., ImageNet, ViT, other high-capacity models). While the new experiments are helpful, they remain limited to relatively modest datasets, and it is still unclear whether the proposed framework can generalize to larger or more realistic scenarios. Given this remaining concern, I am inclined to maintain my original scores.

---

> ### Author Response · Authors · 2025-11-24
> **Real-world Generalizability**
>
> We thank the reviewer for acknowledging that most concerns have been resolved and for raising the remaining issue of scalability.
>
> Seq-CORe50 is widely used in domain incremental learning to evaluate real-world generalizability [1][2], because it provides naturally captured temporal streams with clear domain shifts that match the assumptions of DIL and DIDD. Although its scale is slightly smaller than ImageNet, it remains a standard benchmark for assessing whether a method can cope with realistic domain progressions.
>
> Regarding larger benchmarks such as ImageNet, prior work on dataset distillation reports that distilling even a single ImageNet domain requires substantial computation. Extending such protocols to multiple domains at ImageNet scale is therefore beyond what can be realistically completed within the review period.
> Since our framework is instead designed for offline long-term applications where new domains arrive sequentially, we believe it can be scalable with larger datasets given sufficient computational resources.
>
> To further support this point empirically, we are currently running additional experiments that distill ViT-based models.
> Furthermore, since ImageNet does not provide explicit domain annotations, we also plan to construct ImageNet subsets with artificially defined domains and evaluate DIDD on these subsets.
>
> We hope that this clarification helps alleviate your remaining concerns regarding scalability.
>
> ---
> [1] A Unified Approach to Domain Incremental Learning with Memory: Theory and Algorithm, 2023 NIPS
>
> [2] Dual Consolidation for Pre-Trained Model-Based Domain-Incremental Learning, 2025 CVPR

---

> > ### Author Response · Authors · 2025-11-30
> >
> > To address the scalability concern, we additionally evaluated the performance of the proposed method on six ImageNet-based subsets: ImageNette, ImageWoof, ImageMeow, ImageFruit, ImageYellow, and ImageSquawk, under IPC=1. Since ImageNet does not provide explicit domains, we constructed five domain sequences by selecting five corruption types from ImageNet-C [1] (defocus blur, fog, speckle noise, contrast, and frost). We tested two high-capacity backbone models, ResNet and ViT.
> > The results are summarized in the table below.
> > Although ViT has larger model capacity, its dataset distillation performance is generally lower than that of ResNet, which aligns with the observation in NRR-DD [2] that stronger backbones do not necessarily yield better distillation performance.
> > Nevertheless, the proposed method outperforms the Finetune baseline in most cases and both backbone models, demonstrating that our framework generalizes well to large-scale datasets and modern architectures.
> >
> > |ImageNette|ResNet||ViT||
> > |--------|---------|---------|---------|---------|
> > |Metric| 𝓐ᵀ (↑) | 𝓕ᵀ (↓) | 𝓐ᵀ (↑) | 𝓕ᵀ (↓)
> > | Joint | 40.6|-|18.6|-
> > | Finetune | 32.4|10.0|18.0|9.8
> > |Proposed |42.8|1.3|19.5|4.8
> >
> > |ImageWoof|ResNet||ViT||
> > |--------|---------|---------|---------|---------|
> > |Metric| 𝓐ᵀ (↑) | 𝓕ᵀ (↓) | 𝓐ᵀ (↑) | 𝓕ᵀ (↓)
> > | Joint | 24.2|-|14.4|-
> > | Finetune | 21.6|1.3|12.4|6.3
> > |Proposed |23.0|0.5|15.8|2.3
> >
> > |ImageMeow|ResNet||ViT||
> > |--------|---------|---------|---------|---------|
> > |Metric| 𝓐ᵀ (↑) | 𝓕ᵀ (↓) | 𝓐ᵀ (↑) | 𝓕ᵀ (↓)
> > | Joint | 25.2|-|15.6|-
> > | Finetune |21.0|9.3|14.8|8.5
> > |Proposed |18.2|6.3|15.2|6.8
> >
> > |ImageFruit|ResNet||ViT||
> > |--------|---------|---------|---------|---------|
> > |Metric| 𝓐ᵀ (↑) | 𝓕ᵀ (↓) | 𝓐ᵀ (↑) | 𝓕ᵀ (↓)
> > | Joint | 27.2|-|27.8|-
> > | Finetune | 25.0|4.3|20.2|7.8
> > |Proposed |31.2|1.8|22.2|3.5
> >
> > |ImageYellow|ResNet||ViT||
> > |--------|---------|---------|---------|---------|
> > |Metric| 𝓐ᵀ (↑) | 𝓕ᵀ (↓) | 𝓐ᵀ (↑) | 𝓕ᵀ (↓)
> > | Joint | 50.6|-|30.4
> > | Finetune | 43.0|4.3|20.6|14.5
> > |Proposed |43.4|0.0|28.2|8.0
> >
> > |ImageSquawk|ResNet||ViT||
> > |--------|---------|---------|---------|---------|
> > |Metric| 𝓐ᵀ (↑) | 𝓕ᵀ (↓) | 𝓐ᵀ (↑) | 𝓕ᵀ (↓)
> > | Joint | 40.4|-|26.6|-
> > | Finetune | 36.6|6.0|25.2|2.3
> > |Proposed |42.8|0.5|27.6|1.3
> >
> > ---
> > [1] Benchmarking neural network robustness to common corruptions and perturbations, 2019 ICLR
> >
> > [2] Enhancing dataset distillation via non-critical region refinement, 2025 CVPR

---

### Official Review · Reviewer_1Hqu · 2025-10-28

**Soundness:** 2
**Presentation:** 2
**Contribution:** 2
**Rating:** 4
**Confidence:** 4

**Summary:**

The authors identified that the existing dataset distillation assumes the availability of the entire dataset, while in reality, real datasets are collected incrementally over time. To solve this issue, the authors propose Domain Incremental Dataset Distillation to continuously distill datasets from multiple sources into a single synthetic dataset. Additionally, traditional DD methods sequentially process arriving datasets and cause catastrophic forgetting. The authors propose an Asymmetric Synthetic Data Update strategy to balance the stability-plasticity trade-off:  a bi-level optimization approach based on the meta learning framework to estimate the optimal update rates. The authors perform evaluations to demonstrate the mitigation of the catastrophic forgetting problem and the efficacy of DD in their proposed approach.

**Strengths:**

+ The idea is simple but effective, as demonstrated by the evaluation results that the proposed work achieves the state-of-the-art level performance.
+ The idea is clearly and effectively formulated in mathematical terms, and the writing is very straightforward and easy to follow.

**Weaknesses:**

- While there's beauty in simplicity, the idea is \textit{too} simple. For example, Eq 12 is just a standard gradient-based optimization with the defined losses.
- It seems to me (please correct me if I'm wrong) that $\bar{\alpha}_i$ and $\bar{\beta}_i$ are separately optimized yet they are used jointly (Eq 9). Why aren't they updated jointly instead?
- The theoretical evidence on why this method is working is missing. The existing mathematical formulation is simply descriptive of the approach, not highlighting any insights on why the approach works.
- In a paper that is 9 pages, only 2 pages are the methodology. This balance is a little bit off as I was expecting a longer methodology section.

**Questions:**

Please see "Weaknesses." Mainly I'm concerned why aren't $\bar{\alpha}_i$ and $\bar{\beta}_i$ updated jointly since they are used jointly?

---

> ### Author Response · Authors · 2025-11-20
> **Response for Reviewer 1Hqu**
>
> ### **A1. Joint Optimization of $\alpha$ and $\beta$**
> Thanks for this constructive comment. We apologize for the confusion caused by the simplified equations of updating $\alpha_i$ and $\beta_i$. We clarify that, in our method, $\alpha_i$ and $\beta_i$ are **jointly** optimized to minimize the total meta-objective. Specifically, we optimize the total meta-loss $L_{meta}(\hat{x}) = L_{stable}(\hat{x}) + L_{plastic}(\hat{x}) + L_{penalty}(\hat{x})$ with respect to both $\boldsymbol{\alpha}$ and $\boldsymbol{\beta}$ simultaneously. This ensures that the coefficients are adjusted in a coordinated way to find a global optimum that satisfies the conflicting constraints. As the reviewer pointed out, optimizing them separately leads to suboptimal solutions with degraded performance, as shown in table below. We clarified the joint optimization formulation in the revised manuscript, please see Section 4.2 (Line 251~).
> |R-MNIST|IPC=1|IPC=1|IPC=10|IPC=10|IPC=20|IPC=20|
> |--------|---------|---------|---------|---------|---------|---------|
> |Metric| 𝓐ᵀ (↑) | 𝓕ᵀ (↓) | 𝓐ᵀ (↑) | 𝓕ᵀ (↓) | 𝓐ᵀ (↑) | 𝓕ᵀ (↓) |
> | Separate | 43.3$\pm$1.5 | 37.5$\pm$1.5 | 40.2$\pm$2.2|47.1$\pm$2.0|39.6$\pm$8.1|45.2$\pm$7.8
> | Joint|58.6$\pm$1.3|21.0$\pm$0.8|62.7$\pm$2.0|34.9$\pm$2.0|59.0$\pm$0.5|39.3$\pm$0.4
>
> |Seq-CORe50|IPC=10|IPC=10|IPC=20|IPC=20|
> |--------|---------|---------|---------|---------|
> |Metric| 𝓐ᵀ (↑) | 𝓕ᵀ (↓) | 𝓐ᵀ (↑) | 𝓕ᵀ (↓) |
> | Separate| 51.2$\pm$0.7|35.6$\pm$0.9|54.4$\pm$0.4|36.6$\pm$1.0|
> | Joint| 55.4$\pm$0.7|38.8$\pm$0.5|60.6$\pm$0.4|38.7$\pm$0.3|
>
> |PACS|IPC=10|IPC=10|IPC=20|IPC=20|
> |--------|---------|---------|---------|---------|
> |Metric| 𝓐ᵀ (↑) | 𝓕ᵀ (↓) | 𝓐ᵀ (↑) | 𝓕ᵀ (↓) |
> | Separate | 42.8$\pm$0.4|9.4$\pm$1.4|44.4$\pm$1.0|11.6$\pm$1.4|
> | Joint  | 48.0$\pm$3.1|11.2$\pm$1.8|52.1$\pm$0.6|10.0$\pm$0.5
> ### **A2. Further Insight and Rationale on the Proposed Methodology**
> Thanks for this constructive comment. As the reviewer suggested, we provide deeper analysis of the proposed asymmetric update strategy.
>
> **Penalty terms encourage alpha and beta to act as selective multipliers.**
>
> The meta-loss can be approximated using the first-order Taylor expansion as follows:
> $$
> L_{meta}(\hat{x}\_{meta}) \approx \tilde{L}(\hat{x}) - \eta_\hat{x} \langle \nabla_\hat{x} \tilde{L}(\hat{x}), G(\hat{x}, \alpha, \beta) \rangle,
> $$
> where
> $$
> \tilde{L}(\hat{x}) = L_{stable}(\hat{x}) + L_{plastic}(\hat{x}),
> \quad
> G(\hat{x}, \alpha, \beta) = \sum_{i=1}^N \alpha_i \, g_{S,i} + \beta_i \, g_{P,i}.
> $$
> Thus, a large value of $\alpha_i$ decreases the meta-loss when the inner product
> $\langle \nabla_\hat{x} \tilde{L}(\hat{x}), g_{S,i} \rangle \ge 0$, and similarly for $\beta_i$ when
> $\langle \nabla_\hat{x} \tilde{L}(\hat{x}), g_{P,i} \rangle \ge 0$.
> Our penalty terms encourage $\alpha_i$ and $\beta_i$ to become selective, resulting in sparse coefficients that guide each synthetic sample toward stability or plasticity.
>
> **Theoretical interpretation via KKT conditions.**
>
> We view Domain Incremental Dataset Distillation (DIDD) as a constrained optimization problem in which the synthetic data $\hat{X}$ is required to satisfy stability and plasticity requirements.
> Let $\epsilon_{\mathrm{stable},i}$ and $\epsilon_{\mathrm{plastic},i}$ denote desired tolerance thresholds for each sample $\hat{x}$. Conceptually, the primal problem can be written as
> \begin{align*}
> \text{Find } \hat{x} \quad \text{s.t.} \quad
> L_{stable}({\hat{x}}\_i) \le \epsilon_{stable,i}, \quad
> L_{plastic}({\hat{x}}\_i) \le \epsilon_{plastic,i} \quad \forall i.
> \end{align*}
> Introducing non-negative Lagrange multipliers $\boldsymbol{\alpha}$ and $\boldsymbol{\beta}$, the Lagrangian becomes:
> $$
> \mathcal{L}(\hat{x}, \boldsymbol{\alpha}, \boldsymbol{\beta})
> = \sum_{i=1}^N \left[
> \alpha_i \left(L_{stable}({\hat{x}}\_i) - \epsilon_{stable,i}\right)
> +
> \beta_i \left(L_{plastic}({\hat{x}}\_i) - \epsilon_{plastic,i}\right)
> \right].
> $$
> The gradient of this Lagrangian with respect to $\hat{x}$ matches the asymmetric update direction used in our method, meaning that $\boldsymbol{\alpha}$ and $\boldsymbol{\beta}$ function as the Lagrange multipliers. Our meta-learning algorithm jointly optimizes alpha and beta to minimize the total meta-loss, implicitly approximating the optimal multipliers.
>
> In the ideal constrained problem, the KKT conditions imply the complementary slackness relations
> $$
> \alpha_i \left(L_{stable}({\hat{x}}\_i) - \epsilon_{stable,i}\right) = 0,
> \quad
> \beta_i \left(L_{plastic}({\hat{x}}\_i) - \epsilon_{plastic,i}\right) = 0.
> $$
> This implies that each multiplier becomes non-zero only when its corresponding constraint is active, providing a principled explanation for the asymmetric update strategy. Therefore, the proposed method is not heuristic; it naturally arises as the optimal solution to the underlying constrained optimization problem.
> We incorporated these methodological insights and theoretical evidence in revised manuscript (Section 4.2, Line 275 and 300).

---

### Official Review · Reviewer_sWfL · 2025-10-30

**Soundness:** 2
**Presentation:** 2
**Contribution:** 2
**Rating:** 4
**Confidence:** 4

**Summary:**

The paper proposed a new problem in dataset distillation: Domain Incremental Dataset Distillation (DIDD). The problem assumed that a sequence of domain-shifted datasets (classification datasets that shared the same label space). The difference between this problem and the Continual Learning is the data storage budget is fixed in DIDD. The authors introduce a stability loss to preserve prior knowledge and an asymmetric per-sample update learned via bi-level meta-optimization. The authors conducted experiments to verify their method's effectiveness.

**Strengths:**

1. The authors propose a new problem setting of dataset distilaltion.

2. The proposed method seems address the proposed problem well.

**Weaknesses:**

1. The paper defines a new hybrid setting DIDD by combining dataset distillation and continual learning. However, this formulation appears somewhat artificial and tailored to the proposed method, rather than motivated by a clearly established real-world need or widely recognized benchmark.

2. The paper lacks the baseline of dataset distillation, which condense the accumulated datasets directly. Additionally, the performance comparision is not so fair as the continue learning baselines are not designed for the new DIDD setting.

3. Even in R-MNIST dataset, there is a huge performance loss compared to the whole dataset.

**Questions:**

1. What is the definition of Domain Incremental Dataset Distillation (DIDD)? It was defined as a problem in the abstract, but it was also defined as a framework in contributions.

2. What is the main contribution of this paper? The proposed new problem DIDD? or the new proposed method to address the DIDD problem?

3. Why are there no baselines listed under dataset distillation? Any dataset distillation (DD) method that condenses an accumulated dataset can serve as a valid baseline.

4. How is the computational cost ? There should be an experiments to dicuss the computational cost.

---

> ### Author Response · Authors · 2025-11-20
> **Response for Reviewer sWfL (1/2)**
>
> ### **A1. Problem of DIDD, Main Contributions**
> We appreciate the reviewer for pointing out ambiguous description.
> As defined in Section 4.1, Domain Incremental Dataset Distillation (DIDD) refers to a **problem setting** we introduce. And the term **framework** means the methodology to address the DIDD problem, as shown in Figure 2.
> In this context, our contributions are two-fold: 1) we introduce the new problem of DIDD, and 2) propose a method to effectively address it.
> The problem of DIDD is a necessary and practical formulation, motivated by two fundamental characteristics of real-world applications: 1) limited data storage, and 2) asynchronous access to different datasets (Appendix A.3).
> Whereas existing Dataset Distillation (DD) methods assume the entire dataset is available upfront, which is often impractical, we formulate DIDD to bridge the gap between conventional DD and Continual Learning, addressing the practical needs for handling sequentially arriving domain-shifted data under a fixed storage budget.
> Moreover, to solve the specific challenge of DIDD (i.e., the conflict between stability and plasticity), we proposed the Asymmetric Synthetic Data Update strategy via bi-level optimization.
> This method is the core technical contribution that enables effective distillation in the challenging DIDD scenario, significantly outperforming existing baselines and adapted continual learning methods.
> We clarified this in the revised manuscript.
>
> ### **A2. More DD Baselines**
> As the reviewer suggested, we conducted experiments by employing additional baselines with widely used DD methods: DC and DSA.
> Specifically, we implemented *Joint* and *Gather* methods with DC and DSA whose performance can serve as upper bounds. The comparison results are summarized in the table below. Notably, the proposed method outperforms *Joint* method in some cases, which demonstrates the effectiveness of the proposed method. We are revising Table 1 in the manuscript by adding these results. we revised Table 1 in the manuscript by adding these results.
> | Dataset / IPC | **R-MNIST (1)** | **R-MNIST (10)** | **R-MNIST (20)** | **Seq-CORe50 (10)** | **Seq-CORe50 (20)** | **PACS (10)** | **PACS (20)** |
> |----------------|------------------|-------------------|-------------------|-----------------------|-----------------------|----------------|----------------|
> | Metric         | 𝓐ᵀ (↑)           | 𝓐ᵀ (↑)            | 𝓐ᵀ (↑)            | 𝓐ᵀ (↑)               | 𝓐ᵀ (↑)               | 𝓐ᵀ (↑)        | 𝓐ᵀ (↑)        |
> | **DC (Gather)** | 74.0 $\pm$ 0.4   | 87.8 $\pm$ 0.6  | 94.4 $\pm$ 0.5 |   71.0 $\pm$ 0.8 |  80.1 $\pm$ 0.6 | 50.4 $\pm$ 0.4 | 52.5 $\pm$ 0.3            |
> | **DC (Joint)**  | 53.7 $\pm$ 1.0   | 83.6 $\pm$ 0.4    | 87.4 $\pm$ 0.4    | 34.7 $\pm$ 0.5        | 42.4 $\pm$ 0.9 | 45.3 $\pm$ 0.9 | 48.4 $\pm$ 0.1 |
> | **DSA (Gather)**| 74.4 $\pm$ 1.1   | 92.9 $\pm$ 0.9 | 94.4 $\pm$ 1.0| 71.0 $\pm$ 0.8 | 80.1 $\pm$ 0.6 | 50.2 $\pm$ 1.3 | 54.1$\pm$1.1|
> | **DSA (Joint)** | 55.4 $\pm$ 1.0   | 82.3 $\pm$ 0.4    | 86.4 $\pm$ 0.5    | 33.2 $\pm$ 0.7        | 41.2 $\pm$ 0.7 | 46.1 $\pm$ 1.0 | 48.6 $\pm$ 1.0 |
>
> ### **A3. Limited Performance Gain on R-MNIST**
> Relatively huge performance gap on the R-MNIST dataset is caused by extremely high compression ratios. For instance, when IPC=10, we only maintain **0.008\%, 0.4\% and 0.8\%** of the storage space of the original datasets on the R-MNIST, Seq-CORe50, and PACS datasets, respectively.
> However, it is worth to note that the proposed method consistently outperforms the second-best method across all cases with clear margins~(over about 10\%, 30\%, and 15\% gain of average accuracy on R-MNIST, Seq-CORe50, and PACS datasets, respectively).
> We believe that these significant performance improvements validate the effectiveness of the proposed method.

---

> ### Author Response · Authors · 2025-11-20
> **Response for Reviewer sWfL (2/2)**
>
> ### **A4. Distillation Cost Analysis**
> In the table below, we analyzed the distillation cost of the proposed method compared to that of other baselines, evaluated on the PACS dataset when IPC=10.
> The proposed method incurs a higher distillation cost compared to other baselines due to the additional process of bi-level optimization.
> However, it is important to note that DD is fundamentally an**OFFLINE** procedure executed only **ONCE** prior to model training, and thus does not incur the complexity associated with real-time processing.
> In addition, we conducted an experiment to scale the distillation cost by adjusting the number of iterations as 1,000 during the distillation process.
> Note that, even with a smaller number of iterations, the proposed method (Proposed~(1,000 iter)) still outperforms other baselines significantly, while achieving the lowest distillation cost. We included this result in Section 5.2 (Line 448)  of the revised manuscript.
> | Method                | **P (s)** | **A (s)** | **C (s)** | **S (s)** | **Total (s)** | **𝓐ᵀ (↑)**        | **𝓕ᵀ (↓)**        |
> |-----------------------|-----------|-----------|-----------|-----------|---------------|---------------------|---------------------|
> | **Finetune**          | 112.2     | 102.1     | 101.9     | 94.6      | 410.7         | 26.9 $\pm$ 0.8      | 44.3 $\pm$ 0.3      |
> | **LwF**               | 109.4     | 145.2     | 130.4     | 139.0     | 524.0         | 35.0 $\pm$ 0.6      | 13.0 $\pm$ 0.3      |
> | **EWC**               | 112.5     | 121.9     | 109.8     | 114.1     | 458.3         | 26.4 $\pm$ 0.7      | 43.7 $\pm$ 0.5      |
> | **LF**                | 103.8     | 151.0     | 143.9     | 141.4     | 540.0         | 27.4 $\pm$ 3.4      | 42.9 $\pm$ 2.4      |
> | **MAS**               | 107.5     | 117.3     | 103.6     | 97.5      | 425.8         | 35.0 $\pm$ 1.7      | 7.9 $\pm$ 1.0       |
> | **Proposed**          | 111.4     | 342.5     | 334.2     | 335.1     | 1123.3        | 48.0 $\pm$ 3.1      | 11.2 $\pm$ 1.8      |
> | **Proposed (1000 iter)** | 37.5  | 109.6     | 99.6      | 97.4      | 344.1         | 46.3 $\pm$ 1.8      | 8.5 $\pm$ 1.2       |

---

### Official Review · Reviewer_XAFs · 2025-11-04

**Soundness:** 3
**Presentation:** 3
**Contribution:** 2
**Rating:** 4
**Confidence:** 4

**Summary:**

This paper investigates Domain Incremental Dataset Distillation, where data arrives sequentially. The authors propose Asymmetric Synthetic Data Update, which introduces a stable loss that constrains representation during updates. Additionally, they introduce a meta-learning method and regularization term to balance the weights of the two losses and prevent them from growing excessively.

**Strengths:**

1. The paper is well-written and easy to follow
2. The setting is novel

**Weaknesses:**

1. Meta-learning makes optimization more difficult.
2. Constraining the label space to be the same makes the setting less general.
3. DD is already challenging to train. Adding meta-learning raises concerns about training instability and tuning difficulty.

**Questions:**

1. Does the domain sequence order affect the results?
2. Could you provide non-DD fine-tuning results to help us understand how the gap changes between DD and full incremental domain training?
3. What’s the training cost comparing to baselines?

---

> ### Author Response · Authors · 2025-11-20
> **Response for Reviewer XAFs (1/2)**
>
> ### **A1. Optimization in Meta-Learning Framework**
> Thanks for your insightful comment. As the reviewer pointed out, conventional meta-learning frameworks often encounter optimization difficulties arising from the nested structure of their objectives, where a large number of inner-loop updates may lead to vanishing or exploding gradients. Let $w_{0}$ be the initial parameters that we aim to optimize, $L_{\mathrm{tr}}$ be the training loss for inner-loop updates, and $L_{\mathrm{meta}}$ be the meta loss for outer-loop updates. In the inner loop, the parameters are updated as
> $$w_{t} = w_{t-1} - \eta \nabla_{w_{t-1}} L_{\mathrm{tr}}(w_{t-1}),\quad t = 1,2,\ldots,T,$$
> where $\eta$ is the inner-loop learning rate and $T$ is the number of inner updates. In the outer loop, $w_{0}$ is updated based on the meta loss evaluated at $w_{T}$ as
> $$w_{0} \leftarrow w_{0} - \beta \nabla_{w_{0}} L_{\mathrm{meta}}(w_{T}),$$
> where $\beta$ is the outer-loop learning rate. Using the chain rule, the meta gradient becomes
> $$\nabla_{w_{0}} L_{\mathrm{meta}}(w_{T}) = \nabla_{w_{T}} L_{\mathrm{meta}}(w_{T}) \cdot \prod_{t=1}^{T} \left( I - \eta (\nabla_{w_{t-1}})^{2} L_{\mathrm{tr}}(w_{t-1}) \right).$$
> We can observe that the gradient can either vanish or explode with **increasing** **$T$**. However, the proposed method employs **only a single inner-loop update** ($T = 1$), and thus effectively mitigates these optimization challenges, leading to more stable and efficient training.
> We included this in the revised manuscript. Please see the Appendix A.7.
>
> ### **A2. Label Space Constraint**
> We agree with the reviewer that the label space constraint may limit the applicability of our method in scenarios where tasks have different label spaces.
> However, we would like to note that we first introduce a new problem of **Domain Incremental Dataset Distillation (DIDD)**, where all tasks share the same label space but differ in input distributions, and **Class Incremental Dataset Distillation (CIDD)** with disjoint label spaces is another new problem which is beyond the scope of this paper.
> As we discussed in Appendix, our future research direction includes the application of our methodology to CIDD.
>
> ### **A3. Effect of Sequence Orders**
> Following the reviewer's suggestion, we conducted additional experiments with different orders of training sequences.
> Table below reports the results with randomly generated two different orders (2, 3), in addition to the default order (1), for R-MNIST and PACS datasets. We observe that the proposed method consistently outperforms the baselines with different sequence orders, demonstrating its robustness to the sequence orders.
> We included this in Appendix A.4 of the revised manuscript.
>
> |R-MNIST|Order 1|Order 1|Order 2|Order 2|Order 3|Order 3|
> |--------|---------|---------|---------|---------|---------|---------|
> |Metric| 𝓐ᵀ (↑) | 𝓕ᵀ (↓) | 𝓐ᵀ (↑) | 𝓕ᵀ (↓) | 𝓐ᵀ (↑) | 𝓕ᵀ (↓) |
> | Finetune | 39.4$\pm$2.2 | 59.7$\pm$2.0 | 42.7$\pm$1.7 | 53.6$\pm$1.6 | 40.5$\pm$0.3 | 56.0$\pm$0.4 |
> | MAS | 45.6$\pm$0.9 | 56.3$\pm$0.8 | 46.9$\pm$1.0 | 47.7$\pm$0.9 | 44.2$\pm$0.8 | 50.5$\pm$0.8 |
> | Proposed | 62.7$\pm$2.0 | 34.9$\pm$2.0 | 62.5$\pm$2.0 | 33.3$\pm$2.0 | 63.7$\pm$2.4 | 32.1$\pm$2.4 |
>
> |PACS|Order 1|Order 1|Order 2|Order 2|Order 3|Order 3|
> |--------|---------|---------|---------|---------|---------|---------|
> |Metric| 𝓐ᵀ (↑) | 𝓕ᵀ (↓) | 𝓐ᵀ (↑) | 𝓕ᵀ (↓) | 𝓐ᵀ (↑) | 𝓕ᵀ (↓) |
> | Finetune  | 26.9$\pm$0.8 | 44.3$\pm$0.3 | 35.4$\pm$0.9 | 24.0$\pm$0.6 | 25.7$\pm$2.0 | 32.4$\pm$2.0 |
> | MAS  | 35.0$\pm$1.7 | 7.9$\pm$1.0 | 37.7$\pm$1.0 | 4.9$\pm$1.3 | 40.6$\pm$2.9 | 7.2$\pm$3.4 |
> | Proposed  | 48.0$\pm$3.1 | 11.2$\pm$1.8 | 48.0$\pm$1.2 | 8.2$\pm$1.1 | 50.7$\pm$0.7 | 7.1$\pm$1.4 |

---

> ### Author Response · Authors · 2025-11-20
> **Response for Reviewer XAFs (2/2)**
>
> ### **A4. Non-DD Finetuning Results**
> As the reviewer suggested, we compared the performance of finetuning between DD (dataset distillation) and Non-DD, evaluated on R-MNIST, Seq-CORe50, and PACS datasets. The results are summarized in the below table.
> Non-DD finetuning updates the model directly on the real datasets in a sequential manner, without relying on dataset distillation, and hence consistently achieves higher accuracies than DD finetuning across all datasets.
> | Dataset              | **R-MNIST** | **R-MNIST** | **Seq-CORe50** | **Seq-CORe50** | **PACS** | **PACS** |
> |----------------------|-------------|-------------|----------------|----------------|----------|----------|
> | Metric               | 𝓐ᵀ (↑)      | 𝓕ᵀ (↓)      | 𝓐ᵀ (↑)         | 𝓕ᵀ (↓)         | 𝓐ᵀ (↑)   | 𝓕ᵀ (↓)   |
> | **DD Finetune**      | 41.9 $\pm$ 2.1 | 56.7 $\pm$ 2.1 | 26.4 $\pm$ 0.2 | 79.2 $\pm$ 0.1 | 27.4 $\pm$ 1.2 | 44.4 $\pm$ 1.9 |
> | **Non-DD Finetune**  | 54.1 $\pm$ 2.5 | 47.7 $\pm$ 1.7 | 32.3 $\pm$ 2.6 | 61.7 $\pm$ 0.9 | 37.6 $\pm$ 1.4 | 19.2 $\pm$ 0.3 |
>
> ### **A5. Training Cost**
> In the table below, we analyzed the distillation cost of the proposed method compared to that of other baselines, evaluated on the PACS dataset when IPC=10.
> The proposed method incurs a higher distillation cost compared to other baselines due to the additional process of bi-level optimization.
> However, it is important to note that DD is fundamentally an **OFFLINE** procedure executed only **ONCE** prior to model training, and thus does not incur the complexity associated with real-time processing.
> In addition, we conducted an experiment to scale the distillation cost by adjusting the number of iterations as 1,000 during the distillation process.
> Note that, even with a smaller number of iterations, the proposed method (Proposed~(1,000 iter)) still outperforms other baselines significantly, while achieving the lowest distillation cost.
> We included this result in Section 5.2 (Line 448)  of the revised manuscript.
> | Method                | **P (s)** | **A (s)** | **C (s)** | **S (s)** | **Total (s)** | **𝓐ᵀ (↑)**        | **𝓕ᵀ (↓)**        |
> |-----------------------|-----------|-----------|-----------|-----------|---------------|---------------------|---------------------|
> | **Finetune**          | 112.2     | 102.1     | 101.9     | 94.6      | 410.7         | 26.9 $\pm$ 0.8      | 44.3 $\pm$ 0.3      |
> | **LwF**               | 109.4     | 145.2     | 130.4     | 139.0     | 524.0         | 35.0 $\pm$ 0.6      | 13.0 $\pm$ 0.3      |
> | **EWC**               | 112.5     | 121.9     | 109.8     | 114.1     | 458.3         | 26.4 $\pm$ 0.7      | 43.7 $\pm$ 0.5      |
> | **LF**                | 103.8     | 151.0     | 143.9     | 141.4     | 540.0         | 27.4 $\pm$ 3.4      | 42.9 $\pm$ 2.4      |
> | **MAS**               | 107.5     | 117.3     | 103.6     | 97.5      | 425.8         | 35.0 $\pm$ 1.7      | 7.9 $\pm$ 1.0       |
> | **Proposed**          | 111.4     | 342.5     | 334.2     | 335.1     | 1123.3        | 48.0 $\pm$ 3.1      | 11.2 $\pm$ 1.8      |
> | **Proposed (1000 iter)** | 37.5  | 109.6     | 99.6      | 97.4      | 344.1         | 46.3 $\pm$ 1.8      | 8.5 $\pm$ 1.2       |

---

> > ### Comment · Reviewer_XAFs · 2025-11-22
> >
> > Thanks for the response and the additional results!
> >
> > The 1-step meta-learning claim is reasonable, and the computational cost with 1000 iterations is interesting.
> >
> > I have accordingly revised the score.

---

> > > ### Author Response · Authors · 2025-11-24
> > >
> > > We appreciate your improved evaluation of our work and your constructive feedback.
> > >
> > > By addressing your concerns regarding the optimization difficulty of the meta learning framework and the complexity of the distillation procedure, we believe that the manuscript has become clearer and stronger.
> > >
> > > Please let us know if you have any remaining concerns or points that would benefit from further clarification, as we welcome any suggestions and comments.

---

### Author Response · Authors · 2025-11-20
**General Response to All Reviewers**

We would like to thank all reviewers for their time and constructive feedback. We have done our best to address the concerns raised and provide detailed responses to each point. We are confident that the quality of our manuscript has been significantly strengthened thanks to your insightful comments.

To facilitate the review of our revisions, we have highlighted the major changes and clarifications in the updated manuscript in blue. We remain open to further discussions and welcome any additional constructive feedback.

Best Regards,

Authors

---

### Meta-Review · Area_Chair_Qt74 · 2025-12-30

**Summary:**

This work studies Domain Incremental Dataset Distillation (DIDD), a new problem setting where synthetic data will be updated continuously as new domains arrive, under a fixed memory budget. To mitigate catastrophic forgetting, this work proposed an Asymmetric Synthetic Data Update strategy to balance the stability-plasticity trade-off using meta-learning. Extensive experimental results demonstrate the good performance of the proposed method.

Strength:
1. All reviewers agree that the problem setting of domain incremental dataset distillation is new and interesting.

2. The proposed asymmetric synthetic data update strategy can effectively trade off stability and plasticity.

3. The paper is well written and easy to follow.


Limitations:
1. It may need to present theoretical evidence to justify why this method works well.

2. Some reviewers are a bit concerned about the scalability of the proposed method. The authors did simple experiments on sub-ImageNet datasets to demonstrate its scalability. It would be great to do complete experiments on the whole ImageNet dataset in the revised version.

3. It would be great to do additional experiments to verify that the proposed method can generalize to larger backbones (e.g., ResNet, ViT) or to non-vision modalities.


Overall, the authors have addressed most of the main concerns raised by the reviewers. Thus, I lean towards accepting this paper.

**Reviewer Concerns:**

The authors have addressed the following concerns raised by reviewers.
1. They provide non-DD fine-tuning results.
2. Compare with more baselines in their experiments.
3. Present the computational costs of meta-learning.

**Reviewer Scores:**

Reviewer XAFs tried to raise the score since the authors provide 1-step meta learning to explain the computational costs.

Reviewer 7mfs may raise the score since the authors have addressed all concerns by doing extensive experiments and provide detailed proof.

---

### Decision · Program_Chairs · 2026-01-26

Accept (Poster)